# Molecular dissection of condensin II-mediated chromosome assembly using in vitro assays

**Makoto M Yoshida, Kazuhisa Kinoshita, Yuuki Aizawa, Shoji Tane, Daisuke Yamashita†, Keishi Shintomi, Tatsuya Hirano***

Chromosome Dynamics Laboratory, RIKEN, Wako, Japan

**Abstract** In vertebrates, condensin I and condensin II cooperate to assemble rod-shaped chromosomes during mitosis. Although the mechanism of action and regulation of condensin I have been studied extensively, our corresponding knowledge of condensin II remains very limited. By introducing recombinant condensin II complexes into *Xenopus* egg extracts, we dissect the roles of its individual subunits in chromosome assembly. We find that one of two HEAT subunits, CAP-D3, plays a crucial role in condensin II-mediated assembly of chromosome axes, whereas the other HEAT subunit, CAP-G2, has a very strong negative impact on this process. The structural maintenance of chromosomes ATPase and the basic amino acid clusters of the kleisin subunit CAP-H2 are essential for this process. Deletion of the C-terminal tail of CAP-D3 increases the ability of condensin II to assemble chromosomes and further exposes a hidden function of CAP-G2 in the lateral compaction of chromosomes. Taken together, our results uncover a multilayered regulatory mechanism unique to condensin II, and provide profound implications for the evolution of condensin II.

**\*For correspondence:**
hiranot@riken.jp

**Present address:** †Otsuka Pharmaceutical Co., Ltd, Tokushima, Japan

## Editor's evaluation

This paper examines the specific roles of the condensin II complex subunits in mitotic chromosome condensation in the absence of condensin I in frog egg extracts. The authors found that condensin II subunits influence condensation in a manner that is distinct from condensin I. Since condensin II has additional roles in genome architecture and centromere function, this study will be of high interest to researchers from diverse fields.

## Introduction

Formation of mitotic chromosomes is an indispensable cellular process that ensures the faithful segregation of genomic information to newly forming daughter cells during cell division. During this process, genome DNAs undergo drastic conformational changes and are converted into a set of rod-shaped structures in which sister chromatids are juxtaposed with each other (*Batty and Gerlich, 2019*; *Paulson et al., 2021*). Mis-regulation in these processes often leads to genomic instability and aneuploidy, potentially leading to cancer and birth defects. Deciphering the molecular mechanism of mitotic chromosome assembly is therefore important to understand not only the basic biology of cell proliferation but also the etiology of human diseases that accompany genome instability or chromosome anomalies.

In vertebrates, a pair of large protein complexes, condensin I and condensin II, play critical roles in mitotic chromosome assembly (*Hirano, 2016*; *Uhlmann, 2016*). While both complexes share the same pair of SMC (structural maintenance of chromosomes) ATPase subunits, SMC2 and SMC4, they differ by their kleisin subunit (CAP-H/Brn1 for condensin I and CAP-H2 for condensin II) and

two HEAT-repeat subunits (CAP-D2/Ycs4 and CAP-G/Ycg1 for condensin I; CAP-D3 and CAP-G2 for condensin II). The difference in the subunit composition of the two complexes specifies their spatiotemporal dynamics and functional contributions to mitotic chromosome assembly. Before the onset of mitosis, condensin I is localized to the cytoplasm, whereas condensin II is primarily nuclear (*Hirota et al., 2004*; *Ono et al., 2004*). This spatial separation allows condensin II to initiate structural changes of chromosomes within the prophase nucleus. After nuclear envelope breakdown, condensin I gains access to the chromosomes and completes the formation of rod-shaped structures. Evidence has been provided that condensin II contributes to axial shortening, which is followed by lateral compaction mediated by condensin I (*Green et al., 2012*; *Shintomi and Hirano, 2011*). As judged by an in situ reorganization assay (*Ono et al., 2017*) and a mechanical stretching assay (*Sun et al., 2018*), condensin II plays a more important role in chromosome mechanics than condensin I. An emerging model from high-throughput chromosome conformation capture (Hi-C) and quantitative imaging suggests that condensin II forms large DNA loops, which are in turn subdivided by condensin I into smaller, nested DNA loops (*Gibcus et al., 2018*; *Walther et al., 2018*).

Despite the well-characterized behaviors of condensin I and condensin II in animal tissue culture cells, the relative contribution of condensin I and condensin II to mitotic chromosome assembly is variable among different species and different cell types (*Hirano, 2012*). Condensin I is essential for mitotic chromosome assembly and segregation in all eukaryotic species examined to date. In contrast, condensin II is absent in some species (e.g. fungi including *Saccharomyces cerevisiae*) or non-essential for mitosis in others (e.g. *Drosophila melanogaster* and *Arabidopsis thaliana*). In *Xenopus* egg extracts, condensin I plays a dominant role in mitotic chromosome assembly and condensin II makes a significant, yet rather minor contribution to this process (*Choppakatla et al., 2021*; *Ono et al., 2003*; *Shintomi and Hirano, 2011*). Moreover, a specialized role of condensin II in centromere assembly has been reported in *Xenopus* egg extracts and in human cells (*Barnhart-Dailey et al., 2017*; *Bernad et al., 2011*).

The question of how condensin complexes might mechanistically operate is currently under active investigation. A recent breakthrough in the field was the demonstration that budding yeast condensin (classified as condensin I based on the primary structures of its subunits) has the ability to translocate along a double-stranded DNA (dsDNA) (*Terakawa et al., 2017*) and to 'extrude' a DNA loop (*Ganji et al., 2018*) in an ATP hydrolysis-dependent manner. Structural studies of yeast condensin shed light on the dynamic conformational changes of this highly sophisticated protein machine (*Hassler et al., 2019*; *Kschonsak et al., 2017*; *Lee et al., 2020*). Moreover, a powerful functional assay using *Xenopus* egg extracts revealed intricate functional crosstalk among the individual subunits of mammalian condensin I (*Kinoshita et al., 2015*; *Kinoshita et al., 2022*). In contrast to the rapid accumulation of mechanistic information about condensin I, however, very little progress has been made so far in our understanding of condensin II. Although it has been shown that human condensin I and condensin II display loop extrusion activities with slightly different properties in vitro (*Kong et al., 2020*), it remains unknown how such differences might be related to the different dynamics and functions of the two condensin complexes observed in vivo. Key questions to be addressed are how the individual subunits of condensin II function in chromosome assembly and to what extent the mechanism of action of condensin II is similar to and different from that of condensin I.

In the current study, we have used *Xenopus* egg extracts as a functional assay to address the roles of the individual subunits of the condensin II complex in the initial step of mitotic chromosome assembly. We find that a recombinant condensin II holocomplex can functionally replace endogenous condensin II in the extract lacking condensin I and enables the assembly of 'chenille-like' chromosomes with inner axes. Analyses using subunit-deletion mutant complexes demonstrate that CAP-D3 has a positive role in assembling the axes, whereas CAP-G2 has a very strong negative impact on this process. As opposed to the essential requirements for SMC ATPase and basic amino acid clusters conserved in CAP-H2, the C-terminal tail of CAP-D3 negatively regulates condensin II-mediated chromosome assembly. Interestingly, deletion of the CAP-D3 C-tail can expose a hidden function of CAP-G2 in compacting DNA laterally around the axes. We also show that ATP-stimulated chromatin binding of condensin II can be recapitulated in a simple, extract-free assay. These results not only clarify the mechanistic similarities and differences between condensin I and condensin II but also provide important implications for the evolution of condensin II.

## Results

### Recombinant condensin II can functionally replace endogenous condensin II in *Xenopus* egg extracts

To dissect the functional contributions of the individual subunits of condensin II for mitotic chromosome assembly, we expressed mammalian subunits using a baculovirus expression system in insect cells. A recombinant condensin II holocomplex (holo[WT]) composed of five subunits (*Figure 1A*) was purified according to the procedures described previously (*Kinoshita et al., 2015*; *Kinoshita et al., 2022*) with some modifications (*Figure 1—figure supplement 1A*).

We then tested the ability of condensin II holo(WT) to assemble mitotic chromosomes in *Xenopus* egg extracts. Conventional assays of mitotic chromosome assembly using *Xenopus* egg extracts had used *Xenopus* sperm nuclei as a substrate (*Hirano et al., 1997*; *Ono et al., 2003*; *Shintomi and Hirano, 2011*). More recently, we modified these assays by introducing mouse sperm nuclei into the same extracts, which allowed us to gain additional insights into the mechanism of mitotic chromosome assembly (*Kinoshita et al., 2022*; *Shintomi et al., 2017*). It was noticed that the functional contribution of condensin II was observed more prominently when mouse sperm nuclei were used as a substrate than when *Xenopus* sperm nuclei were used (*Shintomi et al., 2017*). We suspected that the slow kinetics of nucleosome assembly on the mouse sperm substrate creates an environment in favor of condensin II's action. For this reason, mouse sperm nuclei were used as a substrate in the current study. We prepared metaphase extracts depleted of condensin I only (Δcond I) and of condensins I and II together (Δcond I/II) (*Figure 2—figure supplement 1A*). As reported in a previous study (*Shintomi et al., 2017*), control extract (Δmock) produced a cluster of rod-shaped chromosomes whereas condensin I-depleted extract (Δcond I) produced a cluster of thick chromosomes with fuzzy surfaces in which condensin II localized to DAPI-dense axes (*Figure 1B and C*). Immunodepletion of both condensins I and II (Δcond I/II) resulted in the formation of an amorphous mass of chromatin (called 'cloud') in which twenty chromosomal DNAs are entangled with each other. We found that recombinant condensin II holocomplex, when added back into the Δcond I/II extract, effectively restored the ability to support condensin II-mediated chromosome assembly in a dose-dependent manner (*Figure 1B* and *Figure 2—figure supplement 1B*). The addition of holo(WT) at 200 nM caused the formation of chromosomal structures similar to those formed by endogenous condensin II (*Figure 1D and E*). We noticed that this concentration (200 nM) was higher than the concentration of endogenous condensin II (~25 nM) in *Xenopus* egg extracts that had been estimated by proteomic analysis (*Wühr et al., 2014*). Potential explanations for this apparent discrepancy could include the species difference in the primary structures of the subunits or the different states of post-translational modifications. Despite the caveat, these experiments demonstrate that the recombinant mammalian condensin II holocomplex can functionally replace endogenous condensin II in *Xenopus* egg extracts under the current condition. As described here, condensin II, without condensin I, forms a unique chromosome structure with a fuzzy surface and an internal axis. In the current study, we refer to the chromosome structure as a 'chenille-like' chromosome, due to its resemblance to a chenille stem (*Figure 1D*, bottom).

### Chromosome association of condensin II Is decreased by CAP-D3 deletion but is increased by CAP-G2 deletion

Having established that the recombinant condensin II holocomplex works in *Xenopus* egg extracts, we then wished to understand the functional contributions of its HEAT subunits, CAP-D3, and CAP-G2, in the assembly of chenille-like chromosomes. To this end, condensin II subcomplexes lacking CAP-G2 (ΔG2[WT]), CAP-D3 (ΔD3[WT]), or both (ΔD3ΔG2[WT]) (*Figure 1—figure supplement 1A*) were individually added back into extracts depleted of both condensins I and II. Unlike holo(WT), ΔD3(WT) displayed a weak association with chromatin and failed to form DAPI-dense internal axes (*Figure 2A and B*). In contrast, ΔG2(WT) retained the ability to assemble structures reminiscent of chenille-like chromosomes formed by holo(WT) (*Figure 2A and B*). Strikingly, we noticed that an extremely high level of hCAP-H2 signals accumulated on DAPI-dense axes. A time-course of the formation of these chromosomes demonstrated that ΔG2(WT) became detectable on chromatin and initiated the formation of axes at much earlier time points than holo(WT) (*Figure 2—figure supplement 1C*). Unlike ΔG2(WT), ΔD3ΔG2(WT) displayed very little if any activity to associate with chromatin (*Figure 2A and B*).

**Figure 1.** Recombinant condensin II holocomplex can functionally replace endogenous condensin II in *Xenopus* egg extracts. (**A**) A schematic diagram of the condensin II holocomplex that is composed of two structural maintenance of chromosomes (SMC) subunits (SMC2 and SMC4), two HEAT subunits (CAP-D3 and CAP-G2), and a kleisin subunit (CAP-H2). (**B**) Mouse sperm nuclei were incubated with Δmock, Δcond I, or Δcond I/II extracts

*Figure 1 continued on next page*

*Figure 1 continued*

that had been supplemented with a control buffer (buffer) or recombinant (rec) condensin II holocomplex, holo(WT), at a final concentration of 200 nM. After 150 min, reaction mixtures were fixed, split, and labeled with an antibody against either *Xenopus* CAP-H2 (XCAP-H2) or human CAP-H2 (hCAP-H2). DNA was counterstained with DAPI (4′,6-diamidino-2-phenylindole). Shown here is a representative image from over 15 chromosome clusters examined per condition. Scale bar, 10 μm. (**C**) A schematic representation of the chromosome structures observed in Δmock, Δcond I, or Δcond I/II extracts. Rod-shaped chromosomes and chenille-like chromosomes are assembled in Δmock extract and Δcond I extract, respectively. Δcond I/II extract produces an amorphous mass of chromatin ('clouds'). The addition of recombinant condensin II (rec cond II) to the same extract restores its ability to assemble chenille-like chromosomes. (**D**) A single chromosome assembled in Δcond I extract with buffer and Δcond I/II extract with 200 nM holo(WT) from samples from experiment (**B**) in comparison to a chenille stem. Scale bars for chromosomes, 2 μm. (**E**) DAPI intensities from the experiment described in (**B**) were measured along lines drawn perpendicular to chromosome axes to create line scan profiles for the width of chromosomes assembled in the Δcond I extract with buffer and Δcond I/II extract with 200 nM holo(WT) (n=20 chromosomes). The mean and SD were normalized individually to the DAPI intensity (arbitrary unit [a.u.]) at the center of chromosome axes (distance = 0 μm) within each set. A dataset from a single representative experiment out of three repeats is shown.

The online version of this article includes the following source data and figure supplement(s) for figure 1:

**Source data 1.** Microsoft excel of non-normalized data corresponding to *Figure 1E*.

**Figure supplement 1.** Recombinant condensin II complexes used in the study.

**Figure supplement 1—source data 1.** Raw data uncropped gels and blot corresponding to *Figure 1—figure supplement 1*.

**Figure supplement 1—source data 2.** Microsoft excel of DNA constructs used in the study.

---

Having observed progressive accumulation of ΔG2(WT) on chromosome axes over time, the concentration of ΔG2(WT) was titrated down. We found that the addition of as low as 25 nM ΔG2(WT) was sufficient to produce chenille-like chromosomes similar to those formed by the addition of 200 nM holo(WT) (*Figure 2C*). Comparable levels of hCAP-H2 signals were detectable on the axes between the two conditions (*Figure 2D*). Taken together, these results suggest that CAP-D3 and CAP-G2 have a positive and negative role, respectively, in chromosome association of condensin II that leads to chromosome axis formation. A minor concern is that an unidentified residual level of *Xenopus* CAP-G2 remaining in the depleted egg extract may incorporate into the recombinant ΔG2(WT) subcomplex and influence the resultant chromosomal phenotype. However, we consider that this is unlikely because clear functional differences between holo(WT) and ΔG2(WT) are also observed in an extract-free assay, as described below (Figure 6A and B).

## The SMC ATPase cycle is essential for condensin II function

A previous study from our laboratory demonstrated that mutations impairing the SMC ATPase cycle cause very severe defects in condensin I-mediated chromosome assembly in *Xenopus* egg extracts (*Kinoshita et al., 2015*). To test whether the same is true in condensin II-mediated chromosome assembly, we introduced the transition state (TR) mutations, which are predicted to slow down the step of ATP hydrolysis, into the SMC subunits of condensin II (mSMC2 E1114Q and mSMC4 E1218Q), and prepared holo(SMC-TR) and ΔG2(SMC-TR) (*Figure 1—figure supplement 1B*). We found that holo(SMC-TR), when added at 200 nM, bound only weakly to chromatin, leaving an amorphous cloud-like structure (*Figure 3A and B*). At the same concentration, chromosome association of ΔG2(SMC-TR) was greatly reduced compared to that of ΔG2(WT), but was seen at a much higher level than that of holo(WT). Despite the high level of chromosome association, ΔG2(SMC-TR) failed to show any sign of chromosome assembly much like holo(SMC-TR). When ΔG2(WT) and ΔG2(SMC-TR) were reduced to 25 nM, the levels of chromosome association and the resulting chromosome morphology were comparable with those of holo(WT) and holo(SMC-TR), respectively, at 200 nM (*Figure 3A*), a result consistent with that shown in *Figure 2*. These results suggest that ATP hydrolysis facilitates the accumulation of condensin II at the chromosome axes in *Xenopus* egg extracts.

To determine whether CAP-G2 regulates ATP hydrolysis by condensin II, we measured the ATPase activity of these wild-type and mutant complexes. Holo(WT) contained a modest level of ATPase activity in the absence of DNA but was stimulated more than twofold when dsDNA was added (*Figure 3C*). On the other hand, deletion of CAP-G2 resulted in more than threefold increase in



**Figure 2.** Chromosome association of condensin II is decreased by CAP-D3 deletion but is increased by CAP-G2 deletion. (**A**) (Left) Mouse sperm nuclei were incubated with Δcond I/II extracts that had been supplemented with control buffer (buffer), recombinant condensin II holocomplex (holo[WT]) or its subcomplexes (ΔG2[WT], ΔD3[WT], and ΔD3ΔG2[WT]) at 200 nM. After 150 min, reaction mixtures were fixed, and labeled with anti-hCAP-H2 antibody.

*Figure 2 continued on next page*

*Figure 2 continued*

DNA was counterstained with DAPI. The images of the hCAP-H2 signal were captured at different relative exposure times (1 × and 1/4 ×) to show non-saturated signal intensities of chromosome structures produced by ΔG2(WT). Shown here is a representative image from over 15 chromosome clusters examined per condition. Scale bar, 10 μm. (Right) Schematic diagrams of the holocomplex and subcomplexes. (**B**) Signal intensities of hCAP-H2 from the experiment described in (**A**) were divided by DAPI signal intensities and the mean values were normalized relative to the value by holo(WT). The mean ± SD is shown (n=15 chromosome clusters). The p-values listed were assessed by two-tailed Welch's t-test. (**C**) Mouse sperm nuclei were incubated with Δcond I/II extracts that had been supplemented with holo(WT) or ΔG2(WT) at concentrations indicated. After 150 min, reaction mixtures were fixed and processed for immunofluorescence as in (**A**). Shown here is a representative image from over 15 chromosome clusters examined per condition. The right panels show cropped images of the orange sections of the left panels. Scale bar, 10 μm (left) and 5 μm (right). (**D**) DAPI and hCAP-H2 intensities from the experiment described in (**C**) were measured along lines drawn perpendicular to the chromosome to create line scan profiles for the width of chromosomes assembled in Δcond I/II extract with 200 nM holo(WT) or either 25 nM or 200 nM ΔG2(WT) (n=20 chromosomes). The mean and standard deviation were normalized individually to the DAPI intensities (arbitrary unit [a.u.]) at the center of chromosome axes (distance = 0 μm) within each set. Intensities of hCAP-H2 were normalized relative to the central value by 200 nM holo(WT) addition. A dataset from a single representative experiment out of three repeats is shown. WT: wild-type.

The online version of this article includes the following source data and figure supplement(s) for figure 2:

**Source data 1.** Microsoft excel of non-normalized data corresponding to *Figure 2B*.

**Source data 2.** Microsoft excel of non-normalized data corresponding to *Figure 2D*.

**Figure supplement 1.** Immunodepletion of endogenous condensin complexes and basic characterization of recombinant condensin II complexes in *Xenopus* egg extracts.

**Figure supplement 1—source data 1.** Raw data uncropped blots corresponding to *Figure 2—figure supplement 1A*.

---

ATP hydrolysis compared to holo(WT) in the absence of dsDNA. Intriguingly, the ATPase activity of ΔG2(WT) was barely stimulated by adding dsDNA. The introduction of SMC-TR mutations greatly reduced the ATPase activities of both the holocomplex and ΔG2 subcomplex. Taken together, these results demonstrated that CAP-G2 negatively regulates the ATPase activity of condensin II as well as its ability to associate with chromosomes.

## The CAP-D3 C-tail negatively regulates condensin II-mediated chromosome assembly

Human CAP-D3 is a polypeptide with 1498 amino acids, in which a HEAT-repeat-rich section is followed by a C-terminal intrinsically disordered region (IDR) of around 200 amino-acid long (*Figure 4A*; *Ono et al., 2003*; *Yoshimura and Hirano, 2016*). Notably, the C-terminal IDR domain, which we refer to as the C-tail in the current study, contains as many as 11 Cdk1 consensus (S/TP) sites (*Figure 4—figure supplement 1A*). Previous studies reported that mitosis-specific phosphorylation in the CAP-D3 C-tail regulates condensin II functions (*Abe et al., 2011*; *Bakhrebah et al., 2015*).

To address the role of the CAP-D3 C-tail in our functional assay, we prepared holo(D3-dC), a condensin II holocomplex lacking the C-tail (*Figure 1—figure supplement 1C*). When holo(WT) and holo(D3-dC) were incubated in metaphase extracts, the full-length CAP-D3 (D3-FL), but not the C-terminally truncated version of CAP-D3 (D3-dC), underwent progressive, electrophoretic mobility shifts (*Figure 4—figure supplement 1B*). A pair of phosphospecific antibodies against pT1415 and pS1474 recognized shifted D3-FL, but not D3-dC, suggesting that the deletion of the CAP-D3 C-tail removed most, if not all, of the mitosis-specific phosphorylation sites.

We examined the differences between holo(WT) and holo(D3-dC) in two functional assays. In the ATPase assay, we found that the ATPase activity of holo(D3-dC) was significantly higher than that of holo(WT) either in the presence or absence of dsDNA (*Figure 4—figure supplement 1C*). In the add-back assay, higher levels of chromosome association were observed with holo(D3-dC) than with holo(WT) at all concentrations tested (25, 50, and 200 nM; *Figure 4B*). Remarkably, we noticed that the chromosome structure formed by holo(D3-dC) at 200 nM was no longer chenille-like: DAPI signals were more confined toward the axis positive for hCAP-H2 (*Figure 4C*, left, and *Figure 4—figure supplement 2A*), being compacted laterally into structures reminiscent of rod-shaped chromosomes



**Figure 3.** The structural maintenance of chromosome (SMC) ATPase cycle is essential for condensin II function. (**A**) Mouse sperm nuclei were incubated with Δcond I/II extracts that had been supplemented with condensin II holo(WT) and holo(SMC-TR) at 200 nM, or ΔG2(WT) and ΔG2(SMC-TR) at either 200 nM or 25 nM. After 150 min, reaction mixtures were fixed and labeled with anti-hCAP-H2 antibody. DNA was counterstained with DAPI. Shown here is a representative image from over 15 chromosome clusters examined per condition. The images of hCAP-H2 signal were first captured at a single exposure time (1/3 ×) and then increased digitally (1 ×) to visualize signals from chromosomes of complexes apart from ΔG2(WT) at 200 nM. Scale bar, 10 μm. (**B**) Signal intensities of hCAP-H2 from the experiment described in (**A**) were divided by DAPI intensities and the mean values were normalized relative to the value by holo(WT). The mean ± SD is shown (n=15 chromosome clusters). The p-values listed were assessed by two-tailed Welch's t-test. A dataset from a single representative experiment out of two repeats is shown. (**C**) ATPase rates of recombinant condensin II holo(WT), ΔG2(WT),

*Figure 3 continued on next page*

*Figure 3 continued*

holo(SMC-TR), and ΔG2(SMC-TR) in the presence or absence of double-stranded DNA (dsDNA). The error bar represents the SD from three performed repeats. P values listed were assessed by two-tailed t-test. WT: wild-type.

The online version of this article includes the following source data for figure 3:

**Source data 1.** Microsoft excel of non-normalized data corresponding to *Figure 3B*.

**Source data 2.** Microsoft excel of data corresponding to *Figure 3C* and *Figure 4—figure supplement 1C*.

produced by condensin I (*Kinoshita et al., 2022*). However, when the concentration of holo(D3-dC) was lowered to 50 nM, its chromosome association was reduced to a level similar to that of holo(WT) at 200 nM addition and formed similar chenille-like chromosomes (*Figure 4C*, right). These results demonstrated that the CAP-D3 C-tail negatively regulates condensin II-mediated chromosome assembly, and further implicated that a high level of chromosome association of holo(D3-dC) supports the conversion from chenille-like to laterally compacted rod-like structures.

We also questioned what would happen when the deletions of CAP-G2 and the CAP-D3 C-tail were combined. To this end, ΔG2(D3-dC), a ΔG2 subcomplex harboring the deletion of the CAP-D3 C-tail, was prepared and subjected to the add-back assay (*Figure 1—figure supplement 1C*). We found that slightly higher levels of chromosome association were observed with ΔG2(D3-dC) than with ΔG2(WT) (*Figure 4D* and *Figure 4—figure supplement 2A*). Importantly, however, the chromosomes formed by ΔG2(D3-dC), unlike those formed by holo(D3-dC), were chenille-like rather than rod-like, even at the highest concentration of 200 nM (*Figure 4—figure supplement 2B and C*). These results showed that CAP-G2 has a rather cryptic role in chromosome shaping, which can only be uncovered when the holocomplex is overloaded on chromosomes by deleting the CAP-D3 C-tail.

## The basic amino acid clusters of CAP-H2 synergistically contribute to condensin II functions

The kleisin subunits act as the hub of SMC protein complexes including condensin II (*Figure 1A*; *Kong et al., 2020*; *Onn et al., 2007*). A previous study using budding yeast condensin had reported that two basic amino acid clusters (BC1 and BC2) conserved in the central region of Brn1/CAP-H form part of a unique DNA-binding subdomain ('safety belt'; *Kschonsak et al., 2017*). It had also been shown that BC1/2 mutations (substitutions of all basic residues with aspartic acids) in human CAP-H caused hypomorphic defects in condensin I-mediated chromosome assembly in *Xenopus* egg extract (*Kinoshita et al., 2022*).

We wished to test how the corresponding mutations in human CAP-H2 have an impact on condensin II-mediated chromosome assembly in our add-back assay. To this end, we prepared mutant holocomplexes harboring charge-reversal mutations of BC1D, BC2D, or a combination of BC1D and BC2D (BC1/2D) (*Figure 5A* and *Figure 1—figure supplement 1D*). We found that the BC1D or BC2D mutations caused partial defects in chromosome association of condensin II, resulting in the formation of discontinuous axes (*Figure 5B and C*). In contrast, holo(H2-BC1/2D), a holocomplex harboring both BC1D and BC2D mutations, failed to associate with chromatin, leaving a cloud-like morphology. Thus, the two sets of mutations caused synergistic defects in condensin II-mediated chromosome assembly. Interestingly, deletion of CAP-G2 from these mutant complexes harboring BC1D, BC2D, or BC1/2D partially rescued their defects in chromosome association and axis formation (*Figure 5—figure supplement 1*), suggesting that the BC1/2D mutations do not cause complete loss of condensin II functions when the negative regulation by CAP-G2 is eliminated.

## ATP-stimulated chromatin binding of condensin II can be recapitulated in an extract-free assay

The experiments utilizing *Xenopus* egg extracts described above allowed us to dissect the positive and negative roles of the individual subunits in condensin II-mediated chromosome axis assembly. However, the egg extracts contain a number of activities that modify or potentially supplement the activities of the recombinant condensin II complex. In an attempt to fill the gap between the extract-based assay and other biochemical/biophysical assays (*Kong et al., 2020*), we next wished to set up a simple chromatin-binding assay that does not utilize the extracts. A previous study from our laboratory had demonstrated that mitotic chromosome-like structures can be reconstituted by mixing



**Figure 4.** The CAP-D3 C-tail negatively regulates condensin II-mediated chromosome assembly. (**A**) A schematic diagram of the full-length hCAP-D3 (**D3–FL**) and hCAP-D3 lacking the C-terminal tail (**D3–dC**). (**B**) Mouse sperm nuclei were incubated in Δcond I/II extracts that had been supplemented with condensin II holo(WT) or holo(D3-dC) at either 25, 50, or 200 nM. After 150 min, reaction mixtures were fixed and labeled with anti-hCAP-H2 antibody. DNA was counterstained with DAPI. Shown here is a representative image from over 15 chromosome clusters examined per condition. Scale

*Figure 4 continued on next page*

*Figure 4 continued*

bar, 10 µm. (**C**) DAPI and hCAP-H2 intensities from the experiment described in (**B**) were measured along lines drawn perpendicular to chromosome axes to create line scan profiles for the width of chromosomes assembled in Δcond I/II extract with 200 nM holo(WT) or either 50 nM or 200 nM holo(D3-dC) (n=20 chromosomes). The mean and SD were normalized individually to the DAPI intensities (arbitrary unit [a.u.]) at the center of chromosome axes (distance = 0 µm) within each set. Intensities of hCAP-H2 were normalized relative to the central value by 200 nM holo(WT) addition. A dataset from a single representative experiment out of three repeats is shown. (**D**) Mouse sperm nuclei were incubated in Δcond I/II extracts that had been supplemented with condensin II ΔG2(WT) or ΔG2(D3-dC) at either 25, 50, or 200 nM in the same experiment as (**B**). After 150 min, reaction mixtures were fixed and labeled with anti-hCAP-H2 antibody. DNA was counterstained with DAPI. Shown here is a representative image from over 15 chromosome clusters examined per condition. Scale bar, 10 µm. WT: wild-type.

The online version of this article includes the following source data and figure supplement(s) for figure 4:

**Source data 1.** Microsoft excel of non-normalized data of all conditions corresponding to *Figure 4* and *Figure 4—figure supplement 2*.

**Figure supplement 1.** Additional characterization of deletion of the CAP-D3 C-tail.

**Figure supplement 1—source data 1.** Raw data uncropped blots corresponding to *Figure 4—figure supplement 1*.

**Figure supplement 2.** Additional profiles of chromosomes assembled by holo(WT), holo(D3-dC), ΔG2(WT), and ΔG2(D3-dC) from the experiment described in *Figure 4*.

**Figure supplement 3.** Deletion of the CAP-D2 C-tail in condensin I has little impact on chromosome assembly in *Xenopu*s egg extracts.

**Figure supplement 3—source data 1.** Raw data uncropped gel corresponding to *Figure 4—figure supplement 3B*.

**Figure supplement 3—source data 2.** Microsoft excel of non-normalized data corresponding to *Figure 4—figure supplement 3D*.

*Xenopus* sperm nuclei with six purified proteins including condensin I and topoisomerase II (*Shintomi and Hirano, 2021*; *Shintomi et al., 2015*). In this reaction, a core domain of *Xenopus laevis* nucleoplasmin (Npm2) removes sperm-specific proteins from the sperm nuclei to make a 'banana-shaped' swollen chromatin mass.

In the current study, we devised an extract-free assay in which the swollen chromatin mass was used as a simple substrate to test condensin II binding without the influence of factors present in the extracts such as mitosis-specific kinases and chromatin remodelers (*Xenopus* sperm nuclei were used as the substrate in this assay because no reproducible protocol for Npm2-mediated swelling of mouse sperm nuclei was available). In brief, *Xenopus* sperm nuclei were preincubated with Npm2 in the absence or presence of ATP, and the reaction mixtures were supplemented with buffer alone, holo(WT), ΔG2(WT), ΔD3(WT), ΔD3ΔG2(WT), or holo(SMC-TR). Because topoisomerase II was absent in this reaction, no drastic structural change including individualization was expected to occur. We found that condensin II holo(WT) displayed a weak ATP-stimulated binding and caused compaction of the banana-shaped chromatin in an ATP-dependent manner (*Figure 6A and B*). Neither ATP-stimulated binding nor ATP-dependent compaction was observed when the ATP-hydrolysis-deficient holo(SMC-TR) was added. ΔD3(WT) and ΔD3ΔG2(WT) were barely detectable on the chromatin in either the absence or presence of ATP. Strikingly, ΔG2(WT) displayed much greater activities of both ATP-stimulated binding and ATP-dependent compaction than holo(WT). The CAP-D3 C-tail lacking holo(D3-dC) was also tested and was found to display higher activities than holo(WT) (*Figure 6—figure supplement 1*). Thus, the swollen chromatin binding assay allows us to probe for ATP-stimulated chromatin binding and ATP-dependent chromatin compaction by condensin II under an extract-free condition. Importantly, this simple assay could recapitulate, at least in part, the differences in chromatin binding between the wild-type and the mutant complexes observed in *Xenopus* egg extracts.

## Discussion

### Negative regulation of condensin II by CAP-G2 and the CAP-D3 C-tail

In vertebrate cells, condensins I and II have both overlapping and non-overlapping functions and cooperate to assemble mitotic chromosomes for faithful segregation (*Nishide and Hirano, 2014*). One of the primary motivations behind the current study was to clarify the functional similarities and differences between condensin I and condensin II at a mechanistic level. To this end, we have used recombinant condensin II complexes to dissect the functions of its individual subunits in *Xenopus* egg extracts. A previous study from our laboratory had demonstrated that a pair of HEAT subunits of condensin I, CAP-D2, and CAP-G, have antagonistic functions in the dynamic assembly of chromosome axes (*Kinoshita et al., 2015*). Deletion of CAP-G decreased chromosome association of



**Figure 5.** The basic amino acid clusters of CAP-H2 synergistically contribute to condensin II functions. (**A**) A schematic diagram of domain organization of hCAP-H2. CAP-H2 has five motifs that are well conserved among vertebrate species (motifs I to V). Shown here is a sequence alignment of the CAP-H2 orthologs from *Danio rerio* (DrCAP-H2), *Xenopus laevis* (XCAP-H2), *Gallus gallus* (GgCAP-H2), *Mus musculus* (mCAP-H2) and human (hCAP-H2). Conserved residues are labeled in dark blue (Y/F/W), light blue (**P**) and gray (I/M/L*). Within the central region, two basic amino acid clusters (BC1 and BC2 shown in red). The BC1D and BC2D mutations were created by substituting all basic residues (K/R) conserved in BC1 and BC2, respectively, with aspartic acid residues (**D**). The BC1/2D mutations are a combination of the BC1D and BC2D mutations. (**B**) Mouse sperm nuclei were incubated

*Figure 5 continued on next page*

*Figure 5 continued*

with Δcond I/II extracts that had been supplemented with holo(WT), holo(H2-BC1D), holo(H2-BC2D) and holo(H2-BC1/2D) at 200 nM. After 150 min, reaction mixtures were fixed and labeled with anti-hCAP-H2 antibody. DNA was counterstained with DAPI. Shown here is a representative image from over 15 chromosome clusters examined per condition. Scale bar, 10 μm. (**C**) (Left) Signal intensities of hCAP-H2 from the experiment described in (**B**) were divided by DAPI intensities and the mean values were normalized relative to the values by holo(WT). (Right) Signal intensities of hCAP-H2 accumulated at the axes were divided by the overall hCAP-H2 intensities on the chromosomal DNA and the mean values were normalized to the value by holo(WT). The mean ± SD is shown (n=15 chromosome clusters). The p-values listed were assessed by two-tailed Welch's t-test. A dataset from a single representative experiment out of two repeats is shown.

The online version of this article includes the following source data and figure supplement(s) for figure 5:

**Source data 1.** Microsoft excel of non-normalized data corresponding to *Figure 5C*.

**Figure supplement 1.** Deletion of CAP-G2 partially rescues mutations in the basic amino acid clusters of CAP-H2.

**Figure supplement 1—source data 1.** Raw data uncropped gel corresponding to *Figure 5—figure supplement 1*.

condensin I and caused the formation of a chromosome structure with fuzzy surfaces and abnormally thin axes. In striking contrast, deletion of CAP-G2 greatly increased, rather than decreased, chromosome association of condensin II (*Figure 2*). As a consequence, a much lower concentration of ΔG2(WT) (as low as 1/8 of condensin II holo[WT]) was sufficient to assemble chromosomes with chenille-like characteristics. On the other hand, deletion of CAP-D3 decreased chromosome association of condensin II and caused failure to assemble chromosomes with discrete axes. Although the essential requirement of CAP-D3 in condensin II-mediated axis formation was parallel to the requirements of CAP-D2 in condensin I-mediated axis formation (*Kinoshita et al., 2015*; *Kinoshita et al., 2022*), we were surprised to find that CAP-G2 has a very strong negative impact on condensin II functions. Importantly, the results from the ATPase assay (*Figure 3C*) and the extract-free swollen chromatin binding assay (*Figure 6*) also supported the negative role of CAP-G2. We suggest that the balancing acts of the two HEAT subunits, originally proposed for condensin I (*Kinoshita et al., 2015*), also operate in condensin II, but that such actions are fine-tuned differently in each complex (*Figure 7A*). In the case of condensin II, for example, CAP-D3-driven axis assembly is intrinsically dominant, which is counter-balanced by the strong suppressive action of CAP-G2.

Another surprise in the current study was that, like CAP-G2, the CAP-D3 C-tail also plays a negative regulatory role in condensin II functions as judged by multiple functional assays (*Figure 4*, *Figure 4—figure supplement 1C*, and *Figure 6—figure supplement 1*). This mode of regulation is unique to condensin II, because deletion of the CAP-D2 C-tail from the condensin I holocomplex causes no noticeable impact on mitotic chromosome assembly in *Xenopus* egg extracts (*Figure 4—figure supplement 3*; *Kinoshita et al., 2022*). Then, what might be the functional relationship between CAP-G2 and the CAP-D3 C-tail in condensin II regulation? The observation that deletion of the CAP-D3 C-tail has a greater impact on the holocomplex than on the ΔG2 subcomplex (*Figure 4*) suggests an overlap in the negative regulatory roles of CAP-G2 and the CAP-D3 C-tail (*Figure 7A*). A recent study using cross-linking mass spectrometry has revealed physical contacts between an N-terminal region of CAP-G2 and a C-terminal region of CAP-D3 (*Kong et al., 2020*). It is therefore possible that CAP-G2 interferes with the action of CAP-D3 by directly interacting with the CAP-D3 C-tail. Future experiments should test the hypothesis that mitosis-specific phosphorylation by Cdk1 of the CAP-D3 C-tail destabilizes the suppressing interaction, thereby activating CAP-D3-mediated axis assembly.

Despite the potential functional crosstalk between CAP-G2 and the CAP-D3 C-tail, it is important to note that the chromosome structure produced in *Xenopus* egg extract by holo(D3-dC) is qualitatively different from that produced by ΔG2(WT) and ΔG2(D3-dC). While ΔG2(WT) and ΔG2(D3-dC) accumulate at the axis of chromosomes in a dose-dependent manner, such accumulation does not cause enrichment of DNA at the axis (*Figure 4—figure supplement 2B and C*). Instead, ΔG2(WT) and ΔG2(D3-dC) shift the chromosomal DNA to elongate the overall axis while keeping their chenille-like morphology under high-dose conditions (*Figure 7B*). In contrast, holo(D3-dC) increases the DNA level at the axial region under a high-dose condition and produces rod-like chromosomes rather than chenille-like chromosomes. These observations suggest that CAP-G2 is not a mere inhibitor but has a rather cryptic function in the compaction of non-axial bulk chromatin (*Figure 7A*, right). However,

**Figure 6.** ATP-stimulated chromatin binding of condensin II can be recapitulated in an extract-free assay. (**A**) *Xenopus* sperm nuclei were pre-incubated with a buffer containing recombinant nucleoplasmin (Npm2) in the absence or presence of ATP for 30 min, and then holo(WT), ΔG2(WT), ΔD3(WT), ΔD3ΔG2(WT), or holo(TR) was added at 50 nM. After 120 min, reaction mixtures were fixed and labeled with anti-mSMC4 antibody. DNA was counterstained with DAPI. Shown here is a representative image from over 20 nuclei examined per condition. Scale bar, 10 μm. (**B**) (Left) Signal intensities of hCAP-H2 were divided by DAPI intensities and the mean values were normalized relative to the values by holo(WT) without ATP addition (-ATP). (Right) Areas of DAPI signals were measured and the mean values were normalized to values by holo(WT) without ATP addition (-ATP). The mean ± SD is shown (n=20 sperm nuclei). The p-values listed were assessed by two-tailed Welch's t-test. SMC, structural maintenance of chromosomes; WT: wild-type.

*Figure 6 continued on next page*

*Figure 6 continued*

The online version of this article includes the following source data and figure supplement(s) for figure 6:

**Source data 1.** Microsoft excel of non-normalized data corresponding to *Figure 6B*.

**Figure supplement 1.** Characterization of holo(D3-dC) in an extract-free chromatin binding assay.

**Figure supplement 1—source data 1.** Microsoft excel of non-normalized data corresponding to *Figure 6—figure supplement 1B*.

**Figure supplement 1—source data 2.** Microsoft excel of non-normalized data corresponding to *Figure 6—figure supplement 1D*.

this function of CAP-G2 is observed only when the holocomplex is overloaded on chromosomes by deleting the CAP-D3 C-tail. Intriguingly, we find that holo(D3-dC), but not ΔG2(WT), retains DNA-stimulated ATPase activity. Additional comparisons in the future may help further characterize the mechanistic differences between the two mutant complexes.

## How might condensin II work?

Our previous work demonstrated that ATP binding and hydrolysis by the SMC subunits are essential for the functions of condensin I in *Xenopus* egg extracts (*Kinoshita et al., 2015*). When ATP hydrolysis-deficient mutations were introduced into the SMC ATPase subunits, the assembly of axes supported by the condensin II holocomplex or its subcomplex lacking CAP-G2 was completely inhibited (*Figure 3*). Thus, the SMC ATPase cycle is essential for the functions of condensin II, too, even in the absence of its inhibitory subunit CAP-G2.

Two basic patches (BC1 and BC2) present in the central region of budding yeast Brn1/CAP-H, together with Ycg1/CAP-G, have been proposed to form a unique DNA-binding region, referred to as the 'safety belt' (*Kschonsak et al., 2017*). Our previous study demonstrated that charge-reversal mutations (BC1/2D) in the corresponding region of the CAP-H subunit of mammalian condensin I cause 'hyper-compaction' phenotypes in *Xenopus* egg extracts (*Kinoshita et al., 2022*). In contrast, our current results show that the BC1/2D mutations in the CAP-H2 subunit of mammalian condensin II completely abolish its ability to associate with chromatin (*Figure 5*). Although the phenotypic differences observed between condensins I and II are intriguing, it should be noted that there has been no direct evidence to date that mammalian condensins use a safety-belt mechanism for DNA binding as demonstrated by budding yeast condensin. Even if a similar mechanism operates in mammalian condensin II, it is possible that the functional requirement for CAP-G2 is somewhat relaxed, as has been recently shown in condensin from the filamentous fungus *Chaetomium thermophilum* (*Shaltiel et al., 2022*). Moreover, it is worthy to note that ΔG2(H2-BC1/2D), but not holo(H2-BC1/2D), retains the ability to associate with chromatin, suggesting the existence of a loading pathway that is actualized in the absence of CAP-G2 (e.g. CAP-D3-dependent pathway). Future investigation is required to clarify this issue.

It has been proposed that the formation of consecutive loops through a mechanism of loop extrusion underlies mitotic chromosome assembly (*Goloborodko et al., 2016*). A recent single-molecule assay has demonstrated that a human condensin II holocomplex exhibits a loop extrusion activity in vitro, whose properties are slightly different from those of condensin I-mediated loop extrusion (*Kong et al., 2020*). At present, we do not know to what extent the loop extrusion activities measured for the two condensin complexes might directly be coupled to their actions exhibited under physiological conditions. In fact, a recent study from our laboratory has provided evidence that condensin I is equipped with a loop extrusion-independent mechanism that contributes to mitotic chromosome assembly and shaping in *Xenopus* egg extracts (*Kinoshita et al., 2022*). The gap that currently exists between the nanoscale single-molecule assays and the mesoscale chromosome assembly assays needs to be filled (*Birnie and Dekker, 2021*). As an initial step, we have devised an extract-free assay in which an Npm2-mediated swollen chromatin mass is used as a binding substrate for recombinant condensin complexes. This simple assay allowed us to recapitulate ATP-stimulated chromatin binding of condensin II, which in turn results in compaction of the chromatin mass. In the future, multiple functional in vitro assays, such as loop extrusion and reconstitution assays (*Kinoshita et al., 2022*; *Kong et al., 2020*; *Shintomi et al., 2015*), should be set up and combined to assess how condensins work on physiological templates and how their activities are regulated by cell cycle-dependent modifications.

**Figure 7.** Model of condensin II regulation. (**A**) Comparison between condensin I and condensin II. In condensin I, the two HEAT subunits, CAP-D2 and CAP-G, contribute to axis assembly and bulk loading, respectively, and their balancing acts regulate proper chromosome assembly. In condensin II, although CAP-D3 and CAP-G2 contribute to axis assembly and bulk loading, respectively, their balancing acts are modified differently from condensin I: CAP-D3's role in axis assembly is dominant over CAP-G2's role in bulk loading, whereas CAP-G2's function to negatively regulate CAP-D3's function is prominent (bar-headed arrow shown in red). The CAP-D3 C-tail could negatively regulate CAP-D3's action through CAP-G2-dependent (blue arrow shown in blue) and CAP-G2-independent mechanisms (bar-headed arrow shown in blue). (**B**) A schematic representation of the chromosome morphologies formed by holo(WT), ΔG2(WT), and holo(D3-dC) in *Xenopus* egg extracts. Holo(WT) added at 200 nM localizes at the axis and forms chenille-like chromosomes with large loops. ΔG2(WT) and holo(D3-dC) form similar chenille-like chromosomes at lower doses (at 25 nM and 50 nM, respectively). At 200 nM, ΔG2(WT) greatly accumulates at the axis and elongates the axis while keeping the chenille-like morphology. In contrast, holo(D3-dC), when added at 200 nM, also associates with non-axial regions to support lateral compaction, resulting in the formation of a rod-like chromosome.

## Evolutionary implications

The current study has important implications for the evolution of condensin complexes. In contrast to condensin I, which plays an essential role in mitotic chromosome assembly and segregation in most eukaryotes examined so far, the occurrence and use of condensin II are variable among different species (*Hirano, 2012*). For instance, condensin II subunits are not present in fungi including *S. cerevisiae*. The plant *A. thaliana* and the primitive alga *Cyanidioschyzon merolae* possess all three non-SMC subunits of condensin II, but they are non-essential for chromosome assembly and segregation in mitosis (*Fujiwara et al., 2013*; *Sakamoto et al., 2011*). Intriguingly, the *cap-g2* gene is missing in the genome of the fruit fly *D. melanogaster* (*Herzog et al., 2013*), although the *cap-d3* and *cap-h2* genes have meiotic functions and participate in interphase chromosome territory formation (*Hartl et al., 2008*; *Rosin et al., 2018*). Moreover, a recent survey has revealed recurrent losses and rapid evolution of the condensin II complex in insect species (*King et al., 2019*): among the genes encoding the non-SMC subunits of condensin II, *cap-g2* has been lost most frequently during insect evolution. Our current results show that the mutant complex lacking CAP-G2 is able to support chenille-like chromosome assembly even at a lower concentration than the holocomplex. Thus, condensin II can fulfill its functions largely, if not completely, even without the CAP-G2 subunit, providing a possible explanation of why the *cap-g2* gene tends to be lost during evolution in some species.

A recent three-dimensional genomics approach, which applies comprehensive Hi-C analyses to a wide variety of eukaryotic species, has reported that condensin II acts as a determinant of chromosome architectures, namely, Rabl-like or territory-like configuration (*Hoencamp et al., 2021*). The species analyzed in this study included those that possess or lack all condensin II-specific subunits, or those that lack CAP-G2 only. In the future, it will be of interest to understand how evolution has materialized condensin II complexes with different subunit compositions and how such variations have contributed to fine-tuning of chromosome structure and dynamics in the extant species.

# Materials and methods

### Key resources table

| Reagent type (species) or resource | Designation | Source or reference | Identifiers | Additional information |
|---|---|---|---|---|
| Cell line (*Spodoptera frugiperda*) | Sf9 insect cells | ThermoFisher | 11496015 | |
| Cell line (*Spodoptera frugiperda*) | High five insect cells | ThermoFisher | B85502 | |
| Strain, strain background (*Escherichia coli*) | Max efficiency DH10Bac competent cells | ThermoFisher | 10361012 | |
| Strain, strain background (*Escherichia coli*) | BL21(DE3) competent cells | with lab cultured cells | N/A | Electrocompetent cells |
| Biological sample (*Xenopus laevis*) | *Xenopus laevis* egg | Hamamatsu Seibutsu-Kyozai | RRID: NXR 0.031 | Female, adult frogs |
| Biological sample (*Xenopus laevis*) | *Xenopus laevis* sperm nuclei | Hamamatsu Seibutsu-Kyozai | RRID: NXR 0.031 | Male, adult frogs |
| Biological sample (*Mus musculus*) | *Mus musculus* sperm nuclei | *Shintomi et al., 2017*; isolated from *Mus musculus* cauda epididymis (BALB/c×C57BL/6 J)F1 | N/A | Male, adult mouse |
| Antibody | anti-*Xenopus* topoisomerase IIα (rabbit polyclonal serum) | *Hirano and Mitchison, 1994* | in house: αC1-6 | WB (1/2000) |
| Antibody | anti-XCAP-D2 (rabbit polyclonal) | *Hirano et al., 1997* | in house: AfR16L | WB (1.0 µg/mL) |
| Antibody | anti-XCAP-G (rabbit polyclonal) | *Hirano et al., 1997* | in house: AfR11-3L | WB (1.0 µg/mL) |

*Continued on next page*

*Continued*

| Reagent type (species) or resource | Designation | Source or reference | Identifiers | Additional information |
|---|---|---|---|---|
| Antibody | anti-XCAP-H (rabbit polyclonal) | *Hirano et al., 1997* | in house: AfR18 | WB (0.7 µg/mL) |
| Antibody | anti-XSMC2/XCAP-E (rabbit polyclonal) | *Hirano et al., 1997* | in house: AfR9-6 | WB (1.0 µg/mL) |
| Antibody | anti-XSMC4/XCAP-C (rabbit polyclonal) | *Hirano et al., 1997* | in house: AfR8L | WB (2.0 µg/mL) |
| Antibody | anti-mSMC4 (rabbit polyclonal) | *Lee et al., 2011* | in house: AfR326-3L | IF (1.0 µg/mL) |
| Antibody | anti-hCAP-H2 (rabbit polyclonal) | *Ono et al., 2003* | in house: AfR205-4L | WB (0.8 µg/mL); IF (3.0 µg/mL) |
| Antibody | anti-XCAP-D3 (rabbit polyclonal) | *Ono et al., 2003* | in house: AfR196-2L | WB (2.0 µg/mL) |
| Antibody | anti-XCAP-H2 (rabbit polyclonal) | *Ono et al., 2003* | in house: AfR201-4 | WB (2.0 µg/mL) |
| Antibody | anti-XCAP-H2 (rabbit polyclonal) | *Ono et al., 2003* | in house: AfR202-2 | From the same antigen for AfR201; WB (2.0 µg/mL); IF (3.0 µg/mL) |
| Antibody | anti-hCAP-D3 (rabbit polyclonal) | ProteinTech Group | 16828–1-AP; RRID: AB_2282528 | WB (1.0 µg/mL) |
| Antibody | Alexa Fluor 568-conjugated anti-rabbit IgG (goat polyclonal) | ThermoFisher | A11036; RRID: AB_10563566 | IF (1/500) |
| Antibody | anti-hCAP-D3 pS1474 (rabbit polyclonal) | This paper; custom ordered from SIGMA Genosys | in house: AfR358-3P2 | WB (1.0 µg/mL); refer to "Antibodies" subsection |
| Antibody | anti-hCAP-D3 pT1415 (rabbit polyclonal) | This paper; custom ordered from SIGMA Genosys | in house: AfR364-3P2 | WB (1.0 µg/mL); refer to "Antibodies" subsection |
| Antibody | horseradish peroxidase-conjugated anti-rabbit IgG (goat polyclonal) | Vector Laboratories | PI-1000; RRID: AB_2336198 | WB (1/20000) |
| Recombinant DNA reagent | λ DNA | TaKara | 3010 | Used in ATPase assay |
| Sequence-based reagent | hCAP-D3 STOP-to-Glycine forward primer | This paper | N/A | ggccgcgactagttccgttggcagtcttgag |
| Sequence-based reagent | hCAP-D3 STOP-to-Glycine reverse primer | This paper | N/A | ctcaagactgccaacggaactagtcgcggcc |
| Sequence-based reagent | (3 C)-StrepII SpeI/NotI forward oligo | This paper | N/A | ctagtggactggaagttctgttccaggggcccggatcttggagccatccgcaatttgaaaaaggt ggcggttccggcggaggtagcggcggaggttcttggtctcaccctcagttcgagaagtaagc |
| Sequence-based reagent | (3 C)-StrepII SpeI/NotI reverse oligo | This paper | N/A | ggccgcttacttctcgaactgagggtgagaccaagaacctccgccgctacctccgccggaacc gccacctttttcaaattgcggatggctccaagatccgggcccctggaacagaacttccagtcca |
| Sequence-based reagent | StrepII-(3 C) BamHI/EcoRI forward oligo | This paper | N/A | gatccatgtggagccatccgcaatttgaaaaaggtggaggctccggcggaggtagcggcggaggttcttgg tctcaccctcagttcgagaaggaggcggatcactggaagttctgttccagggggcccggggg |
| Sequence-based reagent | StrepII-(3 C) BamHI/EcoRI reverse oligo | This paper | N/A | aattccccgggcccctggaacagaacttccagtgatccgcctcccttctcgaactgagggtgagaccaag aacctccgccgctacctccgccggagcctccacctttttcaaattgcggatggctccacatg |
| Peptide, recombinant protein | Precission Protease | Cytiva | 27-0843-01 | Used in purification of condensins |
| Peptide, recombinant protein | λ protein phosphatase | NEB | P0753 | |
| Peptide, recombinant protein | Benzonase | Novagen | 71205-3CN | Used in purification of condensins |
| Peptide, recombinant protein | BamHI restriction enzyme | TaKaRa | 1010 | |

*Continued on next page*

*Continued*

| Reagent type (species) or resource | Designation | Source or reference | Identifiers | Additional information |
|---|---|---|---|---|
| Peptide, recombinant protein | EcoRI restriction enzyme | TaKaRa | 1040 | |
| Peptide, recombinant protein | NotI restriction enzyme | TaKaRa | 1166 | |
| Peptide, recombinant protein | SpeI restriction enzyme | TaKaRa | 1086 | |
| Peptide, recombinant protein | T4 polynucleotide kinase | TaKaRa | 2021 | |
| Commercial assay or kit | QuikChange II XL Site-Directed Mutagenesis Kit | Agilent Technologies | 200522 | |
| Commercial assay or kit | DNA Ligation Kit | TaKaRa | 6022 | |
| Commercial assay or kit | EnzChek Phosphate Assay Kit | ThermoFisher | E6646 | |
| Software, algorithm | UNICORN 7 | Cytiva | N/A | |
| Software, algorithm | Prism 8 | GraphPad | N/A | |
| Software, algorithm | Excel | Microsoft | N/A | |
| Software, algorithm | Olympus cellSens Dimensions | Olympus | N/A | |
| Software, algorithm | Photoshop | Adobe | N/A | |
| Software, algorithm | ImageJ | *Schneider et al., 2012* | https://imagej.nih.gov/ij/ | |
| Software, algorithm | SparkControl | Tecan Life Sciences | N/A | |
| Other | Coomassie Brilliant Blue R-250 staining solution | Bio-Rad | 1610436 | For polyacrylamide gels |
| Other | Amersham Protran Premium 0.45 nitrocellulose membrane | Cytiva | 10600047 | For Western blots |
| Other | Glutathione Sepharose 4B | Cytiva | 17075601 | Used for purification of condensin I |
| Other | HiTrap Q HP 1 mL | Cytiva | 17115301 | Used for purification of condensins |
| Other | StrepTactin Sepharose HP | Cytiva | 28-9355-99 | Used for purification of condensin II |
| Other | DAPI | Roche | 10236276001 | (2 µg/mL) |
| Other | ATP | Sigma-Aldrich | A2383 | Used in ATPase assay and Npm2-treated chromatin binding assay |
| Other | Dynabeads Protein A | ThermoFisher | 10002D | For immunodepletion with *Xenopus* egg extract |
| Other | Alexa Flour 488-conjugated streptavidin | ThermoFisher | S32354 | IF (1/2000) |

SMC: structural maintenance of chromosomes.

## Antibodies

Primary antibodies used in the current study were as follows: anti-XSMC4/XCAP-C (in-house identifier: AfR8L, affinity-purified rabbit antibody), anti-XSMC2/XCAP-E (AfR9-6, affinity-purified rabbit antibody), anti-XCAP-D2 (AfR16L, affinity-purified rabbit antibody), anti-XCAP-G (AfR11-3L, affinity-purified rabbit antibody), anti-XCAP-H (AfR18-3L, affinity-purified rabbit antibody; *Hirano et al., 1997*; *Hirano and Mitchison, 1994*), anti-XCAP-D3 (AfR196-2L, affinity-purified rabbit antibody), anti-XCAP-H2 (AfR201-4 and AfR202-2, affinity-purified rabbit antibodies; *Ono et al., 2003*), anti-*Xenopus* topoisomerase IIα (in-house identifier: αC1-6, rabbit anti-serum; *Hirano and Mitchison, 1994*), anti-mSMC4 (AfR326-3L, affinity-purified rabbit antibody; *Lee et al., 2011*), anti-hCAP-H2 (AfR205-4L, affinity-purified rabbit antibody; *Ono et al., 2003*), and anti-hCAP-D3 (16828–1-AP

[RRID:AB_2282528], polyclonal rabbit antibody, ProteinTech Group). Custom phospho-specific antibodies against cysteine-included hCAP-D3 pT1415 peptide (C-VTKRAISpTPEK; in-house identifier: AfR364-3P2) and hCAP-D3 pS1474 peptide (C-QQWNVRpSPARNK; in-house identifier: AfR358-3P2) were generated commercially (SIGMA Genosys). Secondary antibodies used in the current study were as follows: horseradish peroxidase-conjugated anti-rabbit IgG (PI-1000 [RRID: AB_2336198], Vector Laboratories, goat antibody), Alexa Fluor 568-conjugated anti-rabbit IgG (A11036 [RRID: AB_10563566], ThermoFisher, goat antibody), and Alexa Fluor 488-conjugated streptavidin (S32354, ThermoFisher).

## Experimental models

Female *Xenopus laevis* frogs (RRID: NXR 0.031, Hamamatsu Seibutsu-Kyozai) were used to lay eggs to harvest *Xenopus* egg extract (*Hirano et al., 1997*). Male *X. laevis* frogs (RRID: NXR 0.031, Hamamatsu Seibutsu-Kyozai) were dissected to prepare sperm nuclei from testes (*Shintomi and Hirano, 2017*). Frogs were used in compliance with the institutional regulations of the RIKEN Wako Campus. Mice (BALB/c×C57BL/6 J)F1) for sperm nuclei (*Shintomi et al., 2017*) were used in compliance with protocols approved by the Animal Care and Use Committee of the University of Tokyo (for M. Ohsugi who provided mouse sperm).

All recombinant condensin complexes were expressed using insect cell strains from the Bacto-Bac Baculovirus Expression System (ThermoFisher) as previously described (*Kinoshita et al., 2015*; *Kinoshita et al., 2022*). Bacmid DNAs were prepared from pFastBac constructs from DH10Bac (10361012, ThermoFisher) according to the manufacturer's instructions and baculoviruses were expressed in Sf9 insect cells (11496015, ThermoFisher) by transfection using Cellfectin II (ThermoFisher) for 7 days before being amplified twice in Sf9 cells for 7 days each. For protein expression, High Five insect cells (B85502, ThermoFisher; 1.2–2.4×10^8 cells) were co-transfected with the recombinant viruses of appropriate combinations and were grown at 28°C for 60–72 hr.

Recombinant *X. laevis* Npm2 was expressed in *Escherichia coli* BL21(DE3) cells as previously described (*Shintomi et al., 2015*).

## Plasmid construction and mutagenesis

Constructs for expressing mSMC2, mSMC4 and their mutant forms were created as previously described (*Kinoshita et al., 2015*). Codon-optimized cDNAs encoding human non-SMC subunits (hCAP-D3, hCAP-G2, and hCAP-H2), based on the amino acid sequence information from *Onn et al., 2007*, were synthesized by ThermoFisher and subcloned into pFastBac1 between *Eco*RI and *Spe*I restriction sites. The following oligonucleotides were used to prepare StrepII-tagged hCAP-D3 and hCAP-H2 DNA constructs: hCAP-D3 STOP-to-Glycine forward primer: 5'-ggccgcgactagttccgttggcagt cttgag-3' hCAP-D3 STOP-to-Glycine reverse primer: 5'-ctcaagactgccaacggaactagtcgcggcc-3'.

(3 C)-StrepII SpeI/NotI forward oligo: 5'-ctagtggactggaagttctgttccaggggcccggatcttggagccatcc gcaatttgaaaaaggtggcggttccggcggaggtagcggcggaggttcttggtctcaccctcagttcgagaagtaagc-3'.

(3 C)-StrepII SpeI/NotI reverse oligo:
5'-ggccgcttacttctcgaactgagggtgagaccaagaacctccgccgctacctccgccggaaccgccacctttttcaaattgcgga tggctccaagatccgggcccctggaacagaacttccagtcca-3'.

StrepII-(3 C) BamHI/EcoRI forward oligo:
5'-gatccatgtggagccatccgcaatttgaaaaaggtggaggctccggcggaggtagcggcggaggttcttggtctcaccctc agttcgagaagggaggcggatcactggaagttctgttccaggggcccgggg-3'.

StrepII-(3 C) BamHI/EcoRI reverse oligo: 5'-aattccccgggcccctggaacagaacttccagtgatccgcctcccttct cgaactgagggtgagaccaagaacctccgccgctacctccgccggagcctccacctttttcaaattgcggatggctccacatg-3'.

To create the hCAP-D3 construct with a C-terminal StrepII-tag (pFastBac1-hCAP-D3-StrepII), the stop codon was first mutagenized to glycine using QuikChange II XL Site-Directed Mutagenesis Kit (Agilent Technologies) with the STOP-to-glycine primers. StrepII oligos with a 3C protease cleavage site within the N-terminal linker sequence were annealed by slow cooling from 95°C, phosphorylated by T4 polynucleotide kinase (TaKaRa) and inserted between *Spe*I and *Not*I restriction sites (pFastBac1-hCAP-D3-[3C]-StrepII) by using restriction enzymes and a DNA ligation kit (TaKaRa). To create the hCAP-H2 construct with an N-terminal StrepII-tag (pFastBac-StrepII-[3C]-hCAP-H2), StrepII oligos with a 3C protease cleavage site within the C-terminal linker sequence were annealed by slow cooling from 95°C, phosphorylated by T4 polynucleotide kinase (TaKaRa), and inserted between *Bam*HI and *Eco*RI

restriction sites by using restriction enzymes and a DNA ligation kit (TaKaRa). Constructs of hCAP-H2 harboring charge-reversal mutations in the BC1 and BC2 regions (H2-BC1D, BC2D, and BC1/2D) were synthesized by ThermoFisher and subcloned into pFastBac1. For the construction of hCAP-D3-dC, a cDNA was synthesized by ThermoFisher so that part of the core amino acid sequence of the 3C protease cleavage site (LFQG) was inserted between hCAP-D3 V1297 and P1298 within the hCAP-D3-StrepII construct. Constructs for recombinant condensin I complexes were prepared as previously described (*Kinoshita et al., 2022*). Human CAP-D2 was used for condensin I holo(WT) while a version that truncates 92 amino acids from the C-terminus (hCAP-D2-dC) was used for condensin I holo(D2-dC).

## Purification of recombinant condensins

For the expression of recombinant condensin complexes, the Bac-to-Bac Baculovirus Expression System (ThermoFisher) was used as described above in the Experimental models subsection. Some modifications were made to the procedure described previously (*Kinoshita et al., 2022*). For recombinant condensin II complexes, baculoviruses used were a combination of mSMC2, GST-mSMC4, hCAP-D3, hCAP-G2, and hCAP-H2. Condensin II wild-type or ATPase TR mutant holocomplexes and ΔG2 subcomplexes, untagged hCAP-D3 was replaced with hCAP-D3-StrepII. For ΔD3 subcomplex and ΔD3ΔG2 subcomplex, untagged hCAP-H2 was replaced with StrepII-hCAP-H2. For complexes that require hCAP-D3-dC, hCAP-D3-StrepII with 3C protease cleavage site insertion between V1297 and P1298 was expressed. Cells expressing condensin II subunits were harvested into pellets, snap-frozen, and stored at −80°C until purification. Cell pellets were resuspended in 20 mL of lysis buffer (10 mM K-HEPES [pH 7.7], 150 mM KCl, 2 mM MgCl$_2$, 0.1 mM CaCl$_2$, 5 mM EGTA, 50 mM sucrose, 0.1% Triton X-100, 1/2 tablet of complete inhibitor [Roche], and 2.5 U/mL Benzonase [Novagen]), lysed by sonication, and clarified by centrifugation. The KCl concentration of the supernatants was adjusted to 300 mM by adding extra KCl and incubated with 2 mL of StrepTactin Sepharose HP beads (Cytiva) at 4°C for 1–2 hr. The beads were then packed into a 12 mL disposable column (Gold Biotechnology), washed with ice-cold purification buffer (10 mM K-HEPES [pH 7.7], 300 mM KCl, 2 mM MgCl$_2$, 0.1 mM CaCl$_2$, 5 mM EGTA, 50 mM sucrose, and 1 mM DTT) four times, and incubated with 5 mL of purification buffer containing 2 mM ATP and 10 mM MgCl$_2$ at room temperature for 15 min twice. After an additional wash with purification buffer, the beads were treated with PreScission Protease (Cytiva) in purification buffer at 4°C overnight to cleave GST and StrepII tags. In the case of holo(D3-dC), PreScission Protease removes the CAP-D3 C-tail by cleaving the 3C site inserted between V1297 and P1298. Cleaved complexes were collected from the column with purification buffer. The eluted fractions were diluted threefold with Buffer A (40 mM Tris [pH 7.5]) to 100 mM KCl and loaded onto a HiTrap Q HP 1 mL column (Cytiva) for ion-exchange chromatography using ÄKTA pure 25 with UNICORN 7 software (Cytiva). Proteins were eluted using a linear gradient from 100 to 1000 mM NaCl with Buffer A and Buffer B (40 mM Tris [pH 7.5] and 1 M NaCl). Fractions containing released condensin II complexes were pooled and dialyzed in condensin II protein buffer (20 mM HEPES [pH 7.7], 150 mM KCl, and 0.5 mM DTT) twice before being concentrated with Amicon Ultra 50 K centrifugal tubes (Merck Millipore). Alternatively, pooled fractions underwent buffer exchange (40 mM Tris [pH 7.5] and 300 mM NaCl) three times using Amicon Ultra tubes before being concentrated (used in *Figure 2A* and *Figure 2— figure supplement 1C*). The stoichiometry of condensin II subunits for each preparation was checked by 5% SDS-PAGE followed by Coomassie Brilliant Blue R-250 (Bio-Rad) staining. Gels were imaged with Amersham Imager 680 (Cytiva). Phosphatase treatment of a purified sample, as described in the Dephosphorylation Assay subsection, caused one of the two bands positive for anti-hCAP-H2 to migrate faster (*Figure 1—figure supplement 1E*), showing that a fraction of hCAP-H2 is phosphorylated in the host insect cells. Expression and purification of recombinant condensin I complexes were performed as previously described (*Kinoshita et al., 2022*). Condensin I holo(WT) was expressed with full-length hCAP-D2 while holo(D2-dC) was expressed with hCAP-D2 lacking 92 amino acids at the C-terminus (hCAP-D2-dC). Condensin I was bound to Glutathione Sepharose 4B beads (Cytiva) in a 5 mL polypropylene column (Qiagen) using GST-tagged mSMC4 and eluted by PreScission Protease (Cytiva) by cleaving the GST-tag. Eluates were collected for ion exchange chromatography as described above. Pooled fractions underwent buffer exchange (40 mM Tris [pH 7.5] and 300 mM NaCl) before being concentrated using Amicon Ultra tubes. The stoichiometry of condensin I subunits for each preparation

was checked by 6% SDS-PAGE (*Figure 4—figure supplement 3B*). Concentrations of recombinant proteins were measured by NanoDrop One[C] (ThermoFisher).

## Dephosphorylation assay

Recombinant condensin II holo(WT) at 500 nM was mixed with or without $\lambda$ protein phosphatase (NEB) at a final concentration of 400 U/µL in 1 x NEBuffer for Protein MetalloPhosphatases supplemented with 1 mM $MnCl_2$ and incubated at 30°C for 60 min. Aliquot samples were taken at 0 and 60 min. Samples were analyzed by immunoblotting. The protein samples were subjected to 7.5% SDS-PAGE and transferred onto a nitrocellulose membrane (Cytiva). The membrane was blocked with 5% skim milk in TBS-T (25 mM Tris [pH 7.5], 0.15 M NaCl, and 0.05% Tween 20) before probing with appropriate affinity-purified rabbit hCAP-H2 antibody in 1% BSA in TBS (25 mM Tris [pH 7.5] and 0.15 M NaCl) followed by horseradish peroxidase-conjugated anti-rabbit IgG (Vector Laboratories) in TBS-T. After washing, the membrane was imaged with Amersham Imager 680 (Cytiva).

## ATPase assay

ATPase assays were performed using the EnzChek Phosphate Assay Kit (Invitrogen). 50 µL reactions contained 200 nM recombinant condensin II in reaction buffer (40 mM Tris [pH 7.5] 15 mM NaCl, 10 mM KCl, 7.5 mM $MgCl_2$, and 2.5 mM ATP), supplemented with 1 × MESG substrate and 1 × purine nucleoside phosphorylase, with or without 1.59 nM (50 µg/µL) bacteriophage $\lambda$ DNA (48,502 bp; TaKaRa). The ratio between condensin II and $\lambda$ DNA was 1 condensin II: 385 bp of DNA. Reactions were placed at 37°C on a 384-well microplate (Greiner Bio-One), and analyzed by spark multimode microplate reader with SparkControl software (Tecan). The initial ATPase rate from timepoint 2.5 min to 7.5 min was determined as ATP molecules hydrolyzed per condensin II complex per min.

## Preparation of *Xenopus* egg extracts

Metaphase high-speed supernatants (M-HSS) of *Xenopus* egg extracts were prepared as previously described (*Hirano et al., 1997*). In brief, *X. laevis* unfertilized eggs were crushed by low-speed centrifugation in XBE2 buffer (10 mM HEPES [pH 7.7], 100 mM KCl, 2 mM $MgCl_2$, 0.1 mM $CaCl_2$, 5 mM EGTA, and 50 mM sucrose). Low-speed supernatant was further fractionated by centrifugation at 200,000 × g for 90 min. The resulting high-speed supernatant was used as a *Xenopus* egg extract for experiments reported in the current study. Immunodepletion was performed using Dynabeads Protein A (ThermoFisher) and DynaMag-2 (Invitrogen) as previously described (*Kinoshita et al., 2022*) with some modifications. To prepare 100 µL of condensin I-depleted (Δcond I) extract, 25 µg of affinity-purified anti-XCAP-D2 were coupled to 100 µL of beads for the first round of depletion, and 25 µg of affinity-purified anti-XCAP-G was coupled to the same vol. of beads for the second round of depletion. To prepare 100 µL of condensins-depleted (Δcond I/II) extract, a mixture of antibodies containing 12.5 µg of affinity-purified anti-XCAP-D2, 6.25 µg affinity-purified anti-XSMC4, and 6.25 µg affinity-purified anti-XSMC2 was coupled to 100 µL of beads for the first round of depletion. Another mixture of antibodies containing 6.25 µg each of affinity-purified anti-XCAP-D3, anti-XSMC2, anti-XCAP-G, and anti-XCAP-H2 (AfR201) was coupled to another 100 µL of beads for the second round of depletion. An extract was incubated with the first beads on ice for 30 min followed by incubation with the second beads on ice for another 30 min. The resulting supernatant was separated from the beads and used as an immunodepleted extract (Δcond I or Δcond II). For mock depletion (Δmock), 25 µg of rabbit IgG (Sigma-Aldrich) was coupled to 100 µL of beads for both the first and second rounds of depletion. The efficiency of immunodepletion was checked by immunoblotting as described above in the Dephosphorylation assay subsection. The membrane was blocked with 5% skim milk in TBS-T before probing with appropriate affinity-purified rabbit antibodies or rabbit serum (anti-XSMC2, XSMC4, XCAP-D2, XCAP-G, XCAP-H, XCAP-D3, XCAP-H2 [AfR201 and AfR202], and XTopo IIα; *Hirano et al., 1997*; *Hirano and Mitchison, 1994*; *Ono et al., 2003*) in 1% BSA in TBS followed by horseradish peroxidase-conjugated anti-rabbit IgG (Vector Laboratories) in TBS-T. After washing, the membrane was imaged with Amersham Imager 680 (Cytiva).

## Mitotic phosphorylation in egg extracts

Mock-depleted (Δmock) or condensins-depleted (Δcond I/II) extracts were supplemented with purified recombinant condensin II complexes or a control condensin II protein buffer and incubated at 22°C

for 60 min. Aliquots were taken at 0, 30, and 60 min and analyzed by immunoblotting as described above in the Dephosphorylation assay subsection. Membranes were probed with rabbit XTopo IIα serum and a rabbit polyclonal anti-hCAP-D3 antibody (ProteinTech Group) in 1% BSA in TBS. When using rabbit anti-phospho-hCAP-D3 antibodies, membranes were blocked with 1% BSA (instead of 5% skim milk) in TBS-T.

## Chromosome assembly in egg extracts

For condensin II-mediated chromosome assembly assays, condensins-depleted (Δcond I/II) extracts were supplemented with purified recombinant condensin II complexes or a control protein buffer and pre-incubated at 22°C for 30 min. Demembranated mouse sperm nuclei were purified as previously described by *Shintomi et al., 2017*. Mouse sperm nuclei were added at a final concentration of about 500–1000 nuclei/µL with an energy-regenerating mixture (1 mM ATP, 10 mM phosphocreatine, and 50 µg/mL creatine kinase) and incubated for another 150 min. The reaction mixtures were fixed by adding 10 × vol. of a fixation solution (10 mM HEPES [pH 7.7], 100 mM KCl, 1 mM MgCl$_2$, and 4% formaldehyde) and incubated at 22°C for 15 min. In the time-course experiments (*Figure 2—figure supplement 1C*), sperm nuclei were added at 0 min and aliquots were taken at time intervals and fixed as above. The fixed samples were loaded onto a 30% glycerol cushion made in 0.5 × XBE2 buffer with a coverslip set inside each tube, and spun down at 5000 rpm at 4°C for 15 min. For condensin I-mediated chromosome assembly assay (*Figure 4—figure supplement 3*), condensins-depleted (Δcond I/II) extracts were supplemented with recombinant condensin I (holo[WT] or holo[D2-dC]) at a final concentration of 35 nM and preincubated at 22°C for 15 min. Mouse sperm nuclei were added and incubated for another 150 min before being treated as described above and as previously described (*Kinoshita et al., 2022*). The coverslips were recovered from the tubes and processed for immunofluorescence labeling as described below in the immunofluorescence labeling subsection.

## Npm2-treated chromatin binding assay

A core domain of *X. laevis* nucleoplasmin (Npm2 [ΔC50]; 1–149 a.a.) was expressed and purified from BL21(DE3) cells as previously described (*Shintomi et al., 2015*). *Xenopus* sperm nuclei, prepared as previously described (*Shintomi and Hirano, 2017*), were incubated with 50 µM Npm2 and 2 mM ATP in KMH buffer (100 mM KCl, 2.5 mM MgCl$_2$, and 20 mM HEPES [pH 7.7]) at 22°C for 30 min. Recombinant condensin II diluted in KMH buffer was added to the mixture at a 1:1 volume so that the final concentrations of condensin II and ATP were 50 nM and 1 mM, respectively, and incubated at 22°C for 120 min. The reaction mixtures were fixed by adding 10 volume of the fixation solution, incubated at 22°C for 15 min, and spun down onto coverslips as described above in the Chromosome assembly in egg extracts subsection. The samples on coverslips were processed for immunofluorescence labeling as described below in the immunofluorescence labeling subsection.

## Immunofluorescence labeling

Immunofluorescence labeling of chromosomes assembled in extracts was performed as described previously (*Kinoshita et al., 2022*) with some modifications. Fixed chromosome samples were spun down onto coverslips, washed with TBS-Tx (25 mM Tris [pH 7.5], 0.15 M NaCl, and 0.1% Triton X-100) three times, and blocked with 1% BSA in TBS-Tx at room temperature for 30–60 min. For immunolabeling, coverslips were incubated with 1% BSA in TBS-Tx containing 3 µg/mL anti-XCAP-H2 (AfR202), anti-hCAP-H2 (AfR205; *Ono et al., 2003*), or biotin-labeled anti-hCAP-H2 at room temperature for 60 min. The coverslips were then washed with TBS-Tx three times and incubated with 1% BSA in TBS-Tx containing a secondary antibody (Alex Fluor 568-conjugated anti-rabbit IgG [1:500; ThermoFisher] or Alexa Fluor 488-conjugated streptavidin [1:2000; ThermoFisher]) at room temperature for 60 min. Finally, the coverslips were incubated with 2 µg/mL DAPI in TBS-Tx for 5 min, washed with TBS-Tx three times, and mounted onto microscope slides with Vectashield H-1000 mounting medium (Vector Laboratories). Immunofluorescence labeling of recombinant condensin I complex was processed as above except with 1 µg/mL rabbit anti-mSMC4 antibody (*Lee et al., 2011*) and Alexa Fluor 568-conjugated anti-rabbit IgG (1:500; ThermoFisher) as previously described (*Kinoshita et al., 2022*).

## Quantification and statistical analysis

Fluorescence microscopy was performed using an Olympus BX63 microscope equipped with a U Plan S-Apo 100 x/1.40 oil immersion lens (Olympus) and ORCA-Flash 4.0 digital CMOS camera C11440 (Hamamatsu Photonics). Olympus cellSens Dimension software (Olympus) was used for image acquisition, and brightness of some image sets in the current study was digitally increased using the exposure setting of Photoshop (Adobe) with the same value to treat each sample within each dataset in the same way. In the case of *Figure 3A*, hCAP-H2 images were first taken at a single time exposure (1/3 ×), and then signals were increased digitally threefolds (1 ×; exposure setting = +2). For colored images (*Figure 1D*), grayscale images were placed in RGB channels using Photoshop (Adobe). ImageJ software (*Schneider et al., 2012*; https://imagej.nih.gov/ij/) was used for quantitative analyses of chromosome and nuclei samples from unedited images. All images used for quantification measurements were captured with identical exposure times that did not cause saturated signals per independent experiment. For quantitative measurements of the signal intensity of DNA, hCAP-H2, and mSMC4 for profiles of chromosome axes, DAPI and immunofluorescence signals were measured on a scanned line perpendicular to the axis at a chromosomal DNA region without DAPI-enriched satellite DNA using the plot profile function as previously described (*Kinoshita et al., 2022*). The three independent experiments mentioned in the legend of *Figure 2D* are separate from the three independent experiments mentioned in legends of *Figure 4* and *Figure 4—figure supplement 2*. For signal intensity quantifications, DAPI and immunofluorescence signals (hCAP-H2) were segmented from the image using the threshold function and the signals in the threshold-selected regions were measured first. The measured immunofluorescence signal (hCAP-H2) was divided by the measured DAPI signal to achieve the hCAP-H2/DAPI signal intensity on chromosomes or swollen chromatin. To calculate the percentage of hCAP-H2 on the axis (hCAP-H2 axis/total intensity) used in *Figure 5C*, the immunofluorescence signal of hCAP-H2 on the axis was segmented from the image using a threshold function with a higher value that only selects the accumulated signal on the chromosome axis and the signal in threshold-selected axis regions were measured. The hCAP-H2 signal at the axis was divided by the overall hCAP-H2 signal on the chromosome to achieve the percentage of hCAP-H2 signal on the axis (hCAP-H2 axis/total intensity). The sample size (cluster of chromatids or swollen chromatin) counted for each experiment is described in the appropriate figure legends. Mean values, SD (error bars), and statistical test values are represented in each figure and described in each figure legend whenever appropriate. All data sets were handled with the Microsoft Excel software (Microsoft) and graphs were created using the GraphPad Prism software (GraphPad).

## Materials availability

All unique/stable reagents generated in this study are available from the corresponding author.

## Acknowledgements

We thank F Inoue, H Watanabe, and M Ohsugi for their help with mouse sperm nuclei preparation and we thank members of the Hirano lab for critically reading the manuscript.

## Additional information

### Competing interests

Daisuke Yamashita: is currently affiliated with Otsuka Pharmaceutical Co., Ltd. The author has no financial interests to declare. The other authors declare that no competing interests exist.

### Funding

| Funder | Grant reference number | Author |
|---|---|---|
| Japan Society for the Promotion of Science | #20K15723 | Makoto M Yoshida |
| Japan Society for the Promotion of Science | #15K06959 | Kazuhisa Kinoshita |

| Funder | Grant reference number | Author |
| --- | --- | --- |
| Japan Society for the Promotion of Science | #19K06499 | Kazuhisa Kinoshita |
| Japan Society for the Promotion of Science | #18H02381 | Keishi Shintomi |
| Japan Society for the Promotion of Science | #19H05755 | Keishi Shintomi |
| Japan Society for the Promotion of Science | #18H05276 | Tatsuya Hirano |
| Japan Society for the Promotion of Science | #20H0593 | Tatsuya Hirano |

The funders had no role in study design, data collection and interpretation, or the decision to submit the work for publication.

## Author contributions

Makoto M Yoshida, Conceptualized the project; Kazuhisa Kinoshita, Contributed to an early stage of the condensin II expression project; Yuuki Aizawa, Performed the ATPase assay and data analysis; Shoji Tane, Performed condensin I purification and the corresponding experiment and data analysis; Daisuke Yamashita, Prepared phosphospecific antibodies against hCAP-D3; Keishi Shintomi, Prepared Npm2 and mouse sperm nuclei; Tatsuya Hirano, Conceptualized the project

## Author ORCIDs

Makoto M Yoshida http://orcid.org/0000-0002-0618-1717
Kazuhisa Kinoshita http://orcid.org/0000-0002-0882-4296
Yuuki Aizawa http://orcid.org/0000-0002-5002-7557
Shoji Tane http://orcid.org/0000-0002-0209-347X
Daisuke Yamashita http://orcid.org/0000-0001-6054-5617
Keishi Shintomi http://orcid.org/0000-0003-0484-9901
Tatsuya Hirano http://orcid.org/0000-0002-4219-6473

## Ethics

Female Xenopus laevis frogs (RRID: NXR 0.031, Hamamatsu Seibutsu-Kyozai) were used to lay eggs to harvest Xenopus egg extract (Hirano et al., 1997). Male X. laevis frogs (RRID: NXR 0.031, Hamamatsu Seibutsu-Kyozai) were dissected to prepare sperm nuclei from testes (Shintomi and Hirano, 2017). Frogs were used in compliance with the institutional regulations of the RIKEN Wako Campus. Mice (BALB/c ×; C57BL/6JF1) for sperm nuclei (Shintomi et al., 2017) were used in compliance with protocols approved by the Animal Care and Use Committee of the University of Tokyo (for M. Ohsugi who provided mouse sperm).

## Decision letter and Author response

Decision letter https://doi.org/10.7554/eLife.78984.sa1
Author response https://doi.org/10.7554/eLife.78984.sa2

# Additional files

## Supplementary files

• MDAR checklist

## Data availability

All data generated or analyzed during this experimental study are included in the manuscript as source data.

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
