## [Editor Report]

This paper examines the specific roles of the condensin II complex subunits in mitotic chromosome condensation in the absence of condensin I in frog egg extracts. The authors found that condensin II subunits influence condensation in a manner that is distinct from condensin I. Since condensin II has additional roles in genome architecture and centromere function, this study will be of high interest to researchers from diverse fields.

---

## [Decision Letter]

**Decision letter after peer review:**

Thank you for submitting your article "Molecular dissection of condensin II-mediated chromosome assembly using in vitro assays" for consideration by *eLife*. Your article has been reviewed by 3 peer reviewers, including Adèle L Marston as Reviewing Editor and Reviewer #1, and the evaluation has been overseen by Jessica Tyler as the Senior Editor. The following individual involved in the review of your submission has agreed to reveal their identity: Alexander E Kelly (Reviewer #2).

Essential revisions:

1) The use of mouse sperm nuclei, instead of *Xenopus* sperm nuclei, for the majority of the analyses is not explained by the authors, and there is a concern that this could be altering or masking phenotypes. In this work, depletion of condensin I results in changes to mouse chromatid morphology, but they are still individualized and there is axially localized condensin. However, in previous papers from the lab (Ono et al., Cell, 2003), depletion of condensin I caused the complete loss of condensation and individualization when *Xenopus* sperm nuclei were used, similar to the depletion of condensin I and II. It is very important that the authors address this directly, as they did in a previous paper (Kinoshita et al., JCB, 2022) where they demonstrated differential effects of certain condensin I mutants depending on the nuclei source. Specifically, the authors should justify the use of mouse sperm in the manuscript in light of their previous findings, and perform key analyses (condensin I depletion, as well as analysis of the G2, D3, and D3 tail mutants) using *Xenopus* sperm. Since it was previously suggested that mouse nuclei are not as dependent on topo II in this system (Shintomi et al., Science, 2017), it might be useful to see if the addition of extra topo II to condensin I-depleted extracts containing *Xenopus* sperm allows for a similar level of condensin-II dependent condensation that is observed with mouse nuclei.

2) More generally, the authors should explain and discuss in more detail their choice to choose to use *Xenopus* extract, mouse and human proteins, and mouse nuclei. How can the authors exclude that the observed effects are not due to the heterologous systems?

3) The depletion of the *Xenopus* CAP-H2 subunit is incomplete in the condI/II depleted extracts (Figure 2—figure supplement 1A). Furthermore, a previous paper from the lab demonstrated that *Xenopus* CAP-G2 is only partially depleted using a similar depletion scheme (Ono et al., Cell, 2003, Figure 5B). A main focus of the work is the role of CAP-G2. However, the depletion of xCAP-G2 from *Xenopus* extracts is not documented (in contrast to many or all other condensin I and condensin II subunits) (Figure 2- supplement 1).

Thus, there are likely significant amounts of CAP-H2 and CAP-G2 in the condI/II depleted extracts, which could be incorporated into the recombinant mouse/human condensin II complexes. In support of this, there is clear axial staining of *Xenopus* CAP-H2 in the "rec cond II holo (WT)" condition, but not the buffer only condition, in condI/II depleted extracts (Figure 1B). This is important, as residual levels of xCAP-G2 may confuse the interpretation of the role of this subunit (the self-inhibition could be explained by an excess of CAP-G2). This could be especially problematic when analyzing the effects of recombinant complexes that do not contain G2 or in which H2 was mutated, especially at lower concentrations of the complexes (see below). The authors should quantify the levels of *Xenopus* CAP-H2 and CAP-G2 on chromatids in all conditions in Figure 1B, as well as provide the levels of mouse SMC2 or SMC4 in the add-back conditions since mouse H2 may not properly reflect the total levels of condensin II on chromatids in the depletion/add-back system. They should also attempt a third round of depletion using anti-XCAP-G and anti-XCAP-H2 antibodies to more completely remove these subunits, and repeat Figures 2C, 4D, and 5B, which are key findings of the paper. If antibodies are not available, proteomics might be suitable to address this point. If this depletion is not technically feasible, the authors should provide clear caveats in the manuscript, and if possible clear up any misunderstandings.

4) Related to points 1 and 3, the concentration of recombinant condensin II complexes in many experiments is 200 nM, including the holo(WT) condition. However, the endogenous concentration of non-SMC condensin II subunits in *Xenopus* egg extracts is much lower (~20-30 nM; Wuhr et al., 2014). Given the dose-dependence effects of the holo(WT) complex on chromatid structure (Figure 2—figure supplement 1B), this would suggest that the mouse/human hybrid complex used here is somehow less active, either due to protein folding issues or evolutionary differences. Alternatively, the high concentration of holo(WT) complexes used here may be required to overcome the loss of a factor that was depleted from extracts that are not present in the purified complex. Further, condensin I complexes are present at ~5 times the concentration of condensin II in egg extracts (Shintomi and Hirano, G and D, 2011), and this difference in relative amounts has been proposed to underlie the reason why condensin II plays a relatively minor role in condensation in egg extracts compared to condensin I. The authors should provide an explanation for why this concentration was used since at 50 nM, which is more similar to endogenous levels of condensin II, holo(WT) was much less effective in promoting chromatid structure when compared to endogenous condensin II (Figure 2—figure supplement 1B vs Figure 1B).

5) Furthermore, it appears that the kinetics of condensation in the ∆cond I/II + condensin II holo(WT) condition (Figure 2—figure supplement 1C) are quite slow compared to WT extracts containing mouse sperm (see Shintomi et al., Science, 2017). The authors should compare the condensation kinetics of WT, cond I -depleted, and ∆cond I/II + condensin II holo(WT) extracts. If cond I-depleted assembly kinetics are more similar to WT than to ∆cond I/II + condensin II holo(WT), this could compromise the validity of the add-back system that forms the basis of the work.

6) It is puzzling that the basic amino acid clusters of CAP-H2, which are supposed to bind DNA and together with CAP-G2 form the safety belt, is essential for DNA binding and certain activity in chromosome condensation, while the CAP-G2 subunit appears dispensable. The authors do not provide an adequate explanation for this finding. Do the basic amino acid clusters act independently of the safety belt and of CAP-G2?

7) Moreover, the H2-BC1/2D mutant complex (lacking the basic residues) is still inhibited by the presence of CAP-G2 (as suggested in Figure 5 —figure supplement). This implies that the basic amino acid clusters are (only?) needed to overcome the inhibition by CAP-G2 but they are not needed for the inhibition itself. This should be more clearly discussed in the manuscript.

8) In the Abstract: 'play positive and negative roles, respectively' is too oversimplified. Depletion experiments globally reveal such roles, but the reality is more complicated with all subunits contributing to the mechanism (positively).

9) Many of the conclusions are backed up by a single representative image. Quantification for all the assays underlying the key conclusions in the manuscript should be presented. In addition, better reporting is required in the figure legends. How many times was each assay performed? How many nuclei were examined in each case?

Please also respond to additional points raised by individual reviewers below.

*Reviewer #1 (Recommendations for the authors):*

1. One of the most interesting findings in the manuscript is the potential relationship between CAP-G2 C terminal region and CAP-D2. The authors speculate that phosphorylation could regulate this relationship. Could the authors test this in their assay?

2. How do the levels of condensin on chromosomes in this assay compare to the levels in vivo? Could the effects observed be due to greatly elevated levels?

*Reviewer #2 (Recommendations for the authors):*

1) Figure 6 doesn't seem to add much new insight, and it is odd that the authors use *Xenopus* sperm in this figure only and don't explain their motivation for this. Thus, I feel it should either be removed from the paper or be moved to supplementary info.

2) The paper would be significantly enhanced with further mechanistic insights into the repressive function of the D3 tail. This could be achieved through phospho-mimetic mutations, and/or through the addition of excess D3 tail to the D3-dC condition to see if it can be inhibited in trans.

3) The cartoon in Figure 1D seems more appropriate for the Discussion and Figure 7. To my knowledge, the authors do not provide direct evidence for the loops and axis featured in 1D, which would require Hi-C experiments.

4) In Figure 1D, for the quantification of the line scans, it would be useful to have chromosomes assembled in WT extracts included so that the reader can be sure of the differences between WT chromosome morphology and "chenille" morphology shown in the representative images are borne out quantitatively.

5) In the ATP-hydrolysis assays in Figure 3, the authors show that ∆G2(WT) mutants have increased ATP activity compared to holo(WT), both with and without dsDNA as a substrate. Given prior work from the lab demonstrating that condensin I goes through continuous cycles of ATP-hydrolysis-dependent turnover in *Xenopus* egg extracts and that transition state mutations slow this turnover rate, it would be interesting to investigate the kinetics of ∆G2(WT) condensin II turnover. If enhanced ATP-hydrolysis were to be a causal factor in the high chromosomal accumulation of condensin II in ∆G2(WT) mutants (which the authors are careful not to claim outright), then we might expect to see an increased rate of condensin II turnover in ∆G2(WT) cond II compared to holo(WT) cond II as well. Such a result would strengthen the possibility of a mechanistic link between CAP-G2's negative regulation of ATP hydrolysis and its negative regulation of condensin II loading on chromosomes. This could be easily tested with the type of two-color assays this lab has previously performed on condensin I mutants.

6) Also in regard to the ATP-hydrolysis assay in Figure 3, it would be worthwhile to include ∆D3(WT) and ∆H2(WT) mutants for completeness. For example, maybe the enhanced ATP-hydrolysis ability of ∆G(WT) isn't specific to CAP-G function, but is instead a result related to sterics, wherein deletion of any of the condensin non-SMC subunits allows more room for ATP to bind and undergo hydrolysis. Demonstrating that ∆D3(WT) does not have increased ATP-hydrolysis activity compared to holo(WT) would eliminate this possibility and further support the notion of CAP-D3 playing a positive role in the chromosomal accumulation of condensin II and axis formation.

7) The authors write that at 200nM, the structures formed by holo(D3-dC) "were no longer chenille-like," but instead "reminiscent of rod-shaped chromosomes produced by condensin I". It does appear that the chromosome axes are better defined in this condition, as confirmed by the line scans in Figure 4C. However, in order to justify the comparison, the authors should include WT chromosomes both in the representative images in 4B and in the line scans in 4C. After all, the difference in the shape of the normalized DAPI intensity distribution (Figure 4C) is not enormous (and there is no statistical analysis of it), so having the WT distribution for comparison would allow the reader to evaluate how complete the rescue actually is. There should be quantification of the ∆G2(WT) vs ∆G2(D3-dC) DAPI images shown in Figure 4D. Figure 4 Figure Supplement 2C has line scans of the relative DAPI intensity distributions of ∆G2(WT) vs WT and ∆G2(D3dC) compared to WT, but the authors need to include the normalized DAPI intensity distributions of ∆G2(WT) vs ∆G(D3-dC) in Figure 4 since these are the conditions being directly compared in Figure 4D and the authors claim that the morphologies are the same at 200nm (i.e. ∆G2(D3-dC) and ∆G2(WT) are both chenille-like). The CAP-H2 intensities should also be included, meaning Figure 4 Figure Supplement 2 B should be moved to Figure 4 (and the 200nm condition should be quantified).

8) In Figure 4 figure supplement 1C, the authors should add ∆G2(D3-dC) mutants to their analysis. ∆G2(D3-dC) mutants are unable to recapitulate the chromosome compaction activity of holo(D3-dC) mutants. It would be natural then, to see if the combination of ∆G2 and D3-dC mutants has any impact on ATP hydrolysis – if combining these mutations lowers the ATP-hydrolysis capacity of the resulting complex, then this would add further insight into why this complex can't match holo(D3-dC) chromosome compaction ability as well as the nature of the "cryptic" function of CAP-G2 exposed by C-terminus deletion.

*Reviewer #3 (Recommendations for the authors):*

1. The quantification of chromosome shape (DNA and condensin distribution across the chromatid) is very useful. However, a direct comparison between conditions incl. wt extracts is difficult as the data is distributed over different graphs. A unifying graph would be helpful.

2. To facilitate comparison across different model systems, please mention ortholog names of CAP-G and -D subunits in the introduction.

3. Abstract: 'how condensin II might work has remained largely unexplored'. I think this statement is not correct (and not needed) as the basic mechanism is understood, but the regulation is much more unexplored.

---

## [Author Response]

Essential revisions:1) The use of mouse sperm nuclei, instead of *Xenopus* sperm nuclei, for the majority of the analyses is not explained by the authors, and there is a concern that this could be altering or masking phenotypes. In this work, depletion of condensin I results in changes to mouse chromatid morphology, but they are still individualized and there is axially localized condensin. However, in previous papers from the lab (Ono et al., Cell, 2003), depletion of condensin I caused the complete loss of condensation and individualization when *Xenopus* sperm nuclei were used, similar to the depletion of condensin I and II. It is very important that the authors address this directly, as they did in a previous paper (Kinoshita et al., JCB, 2022) where they demonstrated differential effects of certain condensin I mutants depending on the nuclei source. Specifically, the authors should justify the use of mouse sperm in the manuscript in light of their previous findings, and perform key analyses (condensin I depletion, as well as analysis of the G2, D3, and D3 tail mutants) using *Xenopus* sperm. Since it was previously suggested that mouse nuclei are not as dependent on topo II in this system (Shintomi et al., Science, 2017), it might be useful to see if the addition of extra topo II to condensin I-depleted extracts containing *Xenopus* sperm allows for a similar level of condensin-II dependent condensation that is observed with mouse nuclei.

As the reviewers correctly point out, we noticed that the functional contribution of condensin II to single chromatid assembly could be observed more readily when mouse sperm nuclei are used as a substrate (Shintomi et al., 2017; the current study). Although the exact reason for this observation is unknown, we suspect that the slow kinetics of nucleosome assembly on the mouse sperm substrate could create an environment in favor of condensin II’s action. We, therefore, reasoned that the use of mouse sperm nuclei as a substrate might provide deep insights into the action of condensin II, which was otherwise rather cryptic when *Xenopus* sperm nuclei are used as a substrate (Ono et al., 2003). Note that one of the goals of the current study is to dissect the role of the individual subunits of condensin II by using *Xenopus* egg extracts as a functional assay. It is, therefore, reasonable to select the best substrate for this specific goal. We must admit that proper justification for the use of the mouse sperm substrate was missing in the original manuscript. In the revised manuscript, we have added a brief explanation as follows:

Lines 128-137:

“Conventional assays of mitotic chromosome assembly using *Xenopus* egg extracts had used *Xenopus* sperm nuclei as a substrate (Hirano et al., 1997; Ono et al., 2003; Shintomi and Hirano, 2011). More recently, we modified these assays by introducing mouse sperm nuclei into the same extracts, which allowed us to gain additional insights into the mechanism of mitotic chromosome assembly (Shintomi et al., 2017; Kinoshita et al., 2022). It was noticed that the functional contribution of condensin II was observed more prominently when mouse sperm nuclei were used as a substrate than when *Xenopus* sperm nuclei were used (Shintomi et al., 2017). We suspected that the slow kinetics of nucleosome assembly on the mouse sperm substrate creates an environment in favor of condensin II’s action. For this reason, mouse sperm nuclei were used as a substrate in the current study.”

2) More generally, the authors should explain and discuss in more detail their choice to choose to use *Xenopus* extract, mouse and human proteins, and mouse nuclei. How can the authors exclude that the observed effects are not due to the heterologous systems?

As far as the source of recombinant condensin subunits is concerned, there is a very long history in the authors’ group. As early as in the mid-1990s, we had attempted to reconstitute recombinant condensin complexes by using *Xenopus* cDNAs as a starting material. This attempt failed because it turned out that the residual expression of a *Xenopus* condensin subunit is highly toxic to bacterial growth and hampered the propagation of its full-length cDNA in bacterial cells. The exact reason for the toxicity was unknown. We then used human cDNAs for the same purpose. Although we were able to report the first reconstitution of condensin holocomplexes from human recombinant subunits (Onn et al., 2007), this trial brought only a partial success: one of the human SMC subunits was poorly expressed in insect cells, making it difficult for us to use them for comprehensive functional assays. We finally overcame these technical difficulties by using mouse cDNAs for the SMC subunits (Kinoshita et al., 2015). Most recently, we have further refined the purification scheme of the mammalian condensin I complexes and reported an exciting set of functional data (Kinoshita et al., 2022). We hope that the reviewers appreciate the fact that biochemical reconstitution of a large protein complex such as condensins remains technically challenging even in this era, and that the current study represents part of a two-decade-long effort of the authors' laboratory. Because the introduction of recombinant mammalian condensin complexes into *Xenopus* egg extracts has been described before (Kinoshita et al., 2015; 2022), we do not think that detailed justification needs to be provided in the current manuscript. We have added brief explanations for the use of mouse sperm nuclei with *Xenopus* egg extract in our response to comment #1 above (Line 127-137).

It should also be noted that our observations made in the combination of mouse sperm nuclei and *Xenopus* egg extracts can be recapitulated, at least in part, with *Xenopus* sperm nuclei in the extract-free assay (Figure 6; Figure 6-fugure supplement 1). Thus, the use of different substrates in experimental setups can be fully justified. Note that we used *Xenopus* sperm nuclei in the extract-free assay because no reproducible protocol for Npm2-mediated swelling of mouse sperm nuclei was available. In the revised manuscript, this technical issue has been briefly mentioned as follows:

Lines 292-294:

“… chromatin remodelers (*Xenopus* sperm nuclei were used as the substrate in this assay because no reproducible protocol for Npm2-mediated swelling of mouse sperm nuclei was available).”

3) The depletion of the *Xenopus* CAP-H2 subunit is incomplete in the condI/II depleted extracts (Figure 2—figure supplement 1A). Furthermore, a previous paper from the lab demonstrated that *Xenopus* CAP-G2 is only partially depleted using a similar depletion scheme (Ono et al., Cell, 2003, Figure 5B). A main focus of the work is the role of CAP-G2. However, the depletion of xCAP-G2 from *Xenopus* extracts is not documented (in contrast to many or all other condensin I and condensin II subunits) (Figure 2- supplement 1).Thus, there are likely significant amounts of CAP-H2 and CAP-G2 in the condI/II depleted extracts, which could be incorporated into the recombinant mouse/human condensin II complexes. In support of this, there is clear axial staining of *Xenopus* CAP-H2 in the "rec cond II holo (WT)" condition, but not the buffer only condition, in condI/II depleted extracts (Figure 1B). This is important, as residual levels of xCAP-G2 may confuse the interpretation of the role of this subunit (the self-inhibition could be explained by an excess of CAP-G2). This could be especially problematic when analyzing the effects of recombinant complexes that do not contain G2 or in which H2 was mutated, especially at lower concentrations of the complexes (see below). The authors should quantify the levels of *Xenopus* CAP-H2 and CAP-G2 on chromatids in all conditions in Figure 1B, as well as provide the levels of mouse SMC2 or SMC4 in the add-back conditions since mouse H2 may not properly reflect the total levels of condensin II on chromatids in the depletion/add-back system. They should also attempt a third round of depletion using anti-XCAP-G and anti-XCAP-H2 antibodies to more completely remove these subunits, and repeat Figures 2C, 4D, and 5B, which are key findings of the paper. If antibodies are not available, proteomics might be suitable to address this point. If this depletion is not technically feasible, the authors should provide clear caveats in the manuscript, and if possible clear up any misunderstandings.

We thank the reviewers for bringing up the issue of the band found in the immunoblot (Figure 2—figure supplement 1A) and immunofluorescence signals detected in the panel in which the anti-XCAP-H2 antibody (in-house identifier AfR201) was used (Figure 1B in the original manuscript). As described below, we now demonstrate that the band found in the lane of the Δcond I/II extract is a non-specific band that migrates very closely to XCAP-H2. In brief, we re-ran immunodepletion samples and tested another XCAP-H2 antibody derived from a different rabbit immunized with the same antigen. This antibody (in-house identifier AfR202) did not detect the non-specific band cross-reacted with AfR201, demonstrating convincingly that XCAP-H2 has been sufficiently immunodepleted in the Δcond I/II sample. To prevent any confusion, we have replaced the original immunoblot set with a new set that includes the results obtained with both XCAP-H2 antibodies (AfR201 and AfR202). We apologize for the potentially misleading presentation made in the original manuscript. In the revised manuscript, we have added the following sentence to the corresponding figure legend of Figure 2—figure supplement 1A:

Lines 1068-1072:

“Immunodepletion of *Xenopus* CAP-H2 (XCAP-H2) was evaluated with two different affinity-purified antibodies (AfR201 and AfR202) raised against the same antigen to confirm that the band migrating slightly slower than XCAP-H2 is a non-specific band cross-reacted only with AfR201 but not with AfR202.”

We then investigated the possible origin of the faint immunofluorescence signal detected as XCAP-H2 (AfR201) on chenille-like chromosomes formed in the Δcond I/II +holo(WT) extract (originally Figure 1B, now moved to Author response image 1). We now provide evidence that the faint signal represents hCAP-H2, rather than XCAP-H2. This problem stems from the double labeling technique using two primary antibodies derived from the same species, as discussed below. For the double labeling of XCAP-H2 and hCAP-H2, we used AfR201 (affinity-purified antibody against XCAP-H2) and biotin-labeled AfR205 (affinity-purified antibody against hCAP-H2), both of which were derived from rabbits. In brief, the samples were incubated with AfR201 and then with Alexa 568-conjugated anti-rabbit IgG to label XCAP-H2. After washing, the samples were incubated with a high concentration of non-immune rabbit IgG to prevent a small, unreacted fraction of the secondary antibody from absorbing any rabbit antibody subsequently added. Following this “quenching” step, the samples were incubated with biotin-labeled AfR205 and then with Alexa 488-conjugated streptavidin to label hCAP-H2. This double labeling method has been well established and used in many previous studies from our laboratory (Kinoshita et al., 2015, 2022; Ono et al., 2003; Shintomi et al., 2011, 2017, 2021; Yamashita et al., 2013). Nevertheless, we notice that the quenching step is sometimes not complete, potentially allowing the Alexa 568-conjugated anti-rabbit IgG to absorb a very small amount of AfR205 subsequently added. If this happened, the channel for Alexa 568 would detect a faint signal that corresponds to hCAP-H2 rather than XCAP-H2.

To rigorously test the possibility discussed above, we first quantified the immunofluorescence signals (divided by DAPI intensity and normalized relative to Δcond I +buffer) presented in the original Figure 1B (Author response image 1). As the reviewers pointed out, a residual yet significant level of signals was measured in the Δcond I/II +holo(WT) extract (Author response image 1). When the double labeling experiment was repeated using AfR202 instead of AfR201, the corresponding signals were still measurable although they were much lower than those detected with AfR201 (Author response image 1 and 1D). Finally, we repeated the experiment and split each sample into two. Each set was then labeled separately with AfR202 or with the anti-hCAP-H2 antibody (single labeling). In this case, a residual signal was no longer detectable with AfR202 in the Δcond I/II +holo(WT) extract (Author response image 1). Taken all together, we conclude that the residual signal detected as XCAP-H2 in the original Figure 1B (Author response image 1) was an artifact derived from incomplete quenching that accompanies the double labeling method. In the revised manuscript, we have decided to present a set of the single labeled images (revised Figure 1B). Accordingly, we have replaced the original panels D and E with those from the same experiment.

The reviewers raise the possibility that XCAP-G2 may remain in the immunodepleted *Xenopus* egg extract and that it may incorporate into the recombinant complex to produce unforeseeable effects. Unfortunately, we no longer have a reliable antibody to probe XCAP-G2 by immunoblotting, but the following arguments make the suggested possibility very unlikely. For instance, the concentration of endogenous XCAP-G2 in *Xenopus* egg extracts was estimated to be ~20-30 nM (Wuhr et al., 2014). If we assume that 5% of it is left in the depleted extract, the final concentration would be ~1-1.5 nM. Note that 25-200 nM of recombinant condensin II were added to the extract, suggesting that only a marginal effect would be expected even when all of the remaining endogenous XCAP-G2 subunit were incorporated into the recombinant complex(es). We also present all possible controls in our add-back assay, which include other mutant complexes such as ΔD3 and ΔD3ΔG2. Most convincingly, the negative role of CAP-G2 is demonstrated also in the extract-free assay using the Npm-treated chromatin as a binding substrate for recombinant condensin II (Figure 6 and Figure 6-supplement figure 1). This assay does not use *Xenopus* egg extract, thereby eliminating the possibility that XCAP-G has an impact, if any, on the action of recombinant condensin II. Moreover, in the revised manuscript, we have included additional results of the chromatin binding assay using holocomplexes harboring hCAP-H2 BC1/2D mutations (Figure 6—figure supplement 1C and 1D). The results also complement those from the *Xenopus* egg extract assay (Figure 5), eliminating the reviewers’ concern that residual XCAP-H might produce unforeseeable effects in the add-back assay. In the revised manuscript, we have modified the text accordingly to emphasize the complementary role of the extract-free assay in the current study:

Lines 281-284:

“However, the egg extracts contain a number of activities that modify or potentially supplement the activities of the recombinant condensin II complex. In an attempt to fill the gap between the extract-based assay and other biochemical/biophysical assays (Kong et al., 2020), we next wished to set up a simple chromatin-binding assay that does not utilize the extracts.”

**Author response image 1. sa2fig1:** Evidence that the faint immunofluorescence signal detected as XCAP-H2 on chenille-like chromosomes formed in the Δcond I/II +holo(WT) extract is an artifact. (A) Double labeling of chromatin/chromosomes with antibodies against XCAP-H2 (AfR201) and hCAP-H2. This panel was presented in Figure 1B in the original manuscript. (B) From the double-labeling experiment described in (A), signal intensities of XCAP-H2 were divided by DAPI signal intensities, and the mean values were normalized relatively to the value by Δcond I extract. The mean ± SD is shown (n = 15 chromosome clusters). Mean values were listed in bold. P values listed were assessed by two-tailed Welch’s t-test. (C) The double-labeling experiment was repeated using antibodies against XCAP-H2 (AfR202) and hCAP-H2. (D) From the double-labeling experiment described in (C), signal intensities of XCAP-H2 were divided by DAPI signal intensities and plotted as above. (E) From the single-labeling experiment shown in the revised Figure 1B, signal intensities of XCAP-H2 were divided by DAPI signal intensities and plotted as above.

4) Related to points 1 and 3, the concentration of recombinant condensin II complexes in many experiments is 200 nM, including the holo(WT) condition. However, the endogenous concentration of non-SMC condensin II subunits in *Xenopus* egg extracts is much lower (~20-30 nM; Wuhr et al., 2014). Given the dose-dependence effects of the holo(WT) complex on chromatid structure (Figure 2—figure supplement 1B), this would suggest that the mouse/human hybrid complex used here is somehow less active, either due to protein folding issues or evolutionary differences. Alternatively, the high concentration of holo(WT) complexes used here may be required to overcome the loss of a factor that was depleted from extracts that are not present in the purified complex. Further, condensin I complexes are present at ~5 times the concentration of condensin II in egg extracts (Shintomi and Hirano, G and D, 2011), and this difference in relative amounts has been proposed to underlie the reason why condensin II plays a relatively minor role in condensation in egg extracts compared to condensin I. The authors should provide an explanation for why this concentration was used since at 50 nM, which is more similar to endogenous levels of condensin II, holo(WT) was much less effective in promoting chromatid structure when compared to endogenous condensin II (Figure 2—figure supplement 1B vs Figure 1B).

We thank the reviewers for bringing up this important issue. It is not fully understood why higher concentrations of recombinant condensin II are required to functionally replace endogenous native condensin II. We consider two possibilities. The first is the species difference in the core activity between *Xenopus* and mammalian condensin II. Although the primary sequences of the condensin II subunits are conserved reasonably well between *Xenopus laevis* and mammals, they are not identical. It is possible that mammalian condensin II does not exhibit its full activity in the environment of *Xenopus* egg extracts. The second possibility is the different states of post-translational modifications. Native condensin II already bears physiological modifications in egg extracts. While we anticipate that the recombinant complexes are modified appropriately and get activated in *Xenopus* egg extracts, there is no guarantee that the anticipated modifications are completed under the current experimental condition. Apart from these issues, we wish to emphasize again that the current work aims to dissect the role of the individual subunits of condensin II by using *Xenopus* egg extracts as a functional assay. In this sense, as long as proper controls are provided, we can draw a solid set of conclusions from this heterologous experimental setup. In the revised manuscript, we have added the following sentences:

Lines 147-156:

“The addition of holo(WT) at 200 nM caused the formation of chromosomal structures similar to those formed by endogenous condensin II (Figure 1D and 1E). We noticed that this concentration (200 nM) was higher than the concentration of endogenous condensin II (~25 nM) in *Xenopus* egg extracts that had been estimated by proteomic analysis (Wuhr et al., 2014). Potential explanations for this apparent discrepancy could include the species difference in the primary structures of the subunits or the different states of post-translational modifications. Despite the caveat, these experiments demonstrate that the recombinant mammalian condensin II holocomplex can functionally replace endogenous condensin II in *Xenopus* egg extracts under the current condition.”

5) Furthermore, it appears that the kinetics of condensation in the ∆cond I/II + condensin II holo(WT) condition (Figure 2—figure supplement 1C) are quite slow compared to WT extracts containing mouse sperm (see Shintomi et al., Science, 2017). The authors should compare the condensation kinetics of WT, cond I -depleted, and ∆cond I/II + condensin II holo(WT) extracts. If cond I-depleted assembly kinetics are more similar to WT than to ∆cond I/II + condensin II holo(WT), this could compromise the validity of the add-back system that forms the basis of the work.

It seems true that the kinetics of chromosome assembly by the recombinant condensin II holo(WT) is slower than that observed in unperturbed *Xenopus* egg extracts. Although the exact reason for this is unknown, we can speculate a number of possible factors that possibly cause the slow kinetics, as discussed above in our response to comment #4. It should also be mentioned that our previous studies using a similar heterologous system produced a wealth of valuable information about the mechanism of action of condensin I (Kinoshita et al., 2015, 2022). At present, no other experimental systems enable the functional dissection of multiprotein complexes under physiological conditions at this level of precision. In the revised manuscript, we have added a brief sentence that describes the limitation of the current approach (Lines xxx-xxx), as mentioned above in our response to comment #4.

6) It is puzzling that the basic amino acid clusters of CAP-H2, which are supposed to bind DNA and together with CAP-G2 form the safety belt, is essential for DNA binding and certain activity in chromosome condensation, while the CAP-G2 subunit appears dispensable. The authors do not provide an adequate explanation for this finding. Do the basic amino acid clusters act independently of the safety belt and of CAP-G2?

The safety belt mechanism composed of Brn1/CAP-H and Ycg1/CAP-G has been proposed based on experiments using yeast condensin, which is closer to condensin I than condensin II in mammals. It is still unknown whether mammalian condensins also function through a similar mechanism, or whether the safety belt mechanism is specific to condensin I or is also applied to condensin II. Although we show that the BC1/2D mutations in CAP-H2, but not loss of CAP-G2, cause complete loss of activities, the currently available evidence is insufficient to discuss this issue beyond the following statements placed in the Discussion of the original manuscript:

Line 385:

“it should be noted that there has been no direct evidence to date that mammalian condensins use a safety-belt mechanism for DNA binding as demonstrated by budding yeast condensin. Further investigation is required to clarify whether mammalian condensins operate through a safety belt mechanism.”

Nevertheless, we have noticed a recent paper reporting that the Ycg1/CAP-G may not be an essential component in the proposed safety belt in the filamentous fungus *Chaetomium thermophilum* (Shaltiel et al., 2022). In the revised manuscript, we have added the following sentence to clarify this issue:

Lines 388-390:

“Even if a similar mechanism operates in mammalian condensin II, it is possible that the functional requirement for CAP-G2 is somewhat relaxed, as has been recently shown in condensin from the filamentous fungus *Chaetomium thermophilum* (Shaltiel et al., 2022).”

7) Moreover, the H2-BC1/2D mutant complex (lacking the basic residues) is still inhibited by the presence of CAP-G2 (as suggested in Figure 5 —figure supplement). This implies that the basic amino acid clusters are (only?) needed to overcome the inhibition by CAP-G2 but they are not needed for the inhibition itself. This should be more clearly discussed in the manuscript.

One possibility is that an additional chromatin targeting region(s) is present in the condensin II complex but is physically hidden or functionally suppressed by the presence of CAP-G2. Along this line, our previous studies provided evidence that mammalian condensin I has a loading mechanism that is independent of the proposed safety belt and that CAP-D2 is involved in such a mechanism (Hara et al., 2019; Kinoshita et al., 2022). The current study shows that the action of CAP-D3 is negatively regulated by CAP-G2, supporting the notion that CAP-D3 is responsible for condensin II loading in the context of DG2(H2-BC1/2D). To make this point clearer, we have added the following brief sentence in the revised manuscript:

Lines 390-393:

“Moreover, it is worthy to note that ΔG2(H2-BC1/2D), but not holo(H2-BC1/2D), retains the ability to associate with chromatin, suggesting the existence of a loading pathway that is actualized in the absence of CAP-G2 (e.g., CAP-D3-dependent pathway).”

8) In the Abstract: 'play positive and negative roles, respectively' is too oversimplified. Depletion experiments globally reveal such roles, but the reality is more complicated with all subunits contributing to the mechanism (positively).

We agree with this comment. We have modified the sentence so that it focuses on the role of the two HEAT subunits in the loading of condensin II and its impact on chromosome axis assembly (rather than chromosome assembly in general) as follows:

Lines 34-36:

“We find that one of two HEAT subunits, CAP-D3, plays a crucial role in condensin II-mediated assembly of chromosome axes whereas the other HEAT subunit, CAP-G2, has a very strong negative impact on this process.”

9) Many of the conclusions are backed up by a single representative image. Quantification for all the assays underlying the key conclusions in the manuscript should be presented. In addition, better reporting is required in the figure legends. How many times was each assay performed? How many nuclei were examined in each case?

We are somewhat confused with this comment. In the original manuscript, we included the quantification (ie., line scans, signal intensity, and/or nuclei size measurements) in the figure panels along with the representative images, as well as the number of experimental repeats and the number of chromosome clusters or nuclei measured (n value), wherever applicable. In the revised manuscript, however, we have made some modifications in the figure legends to make the link between the image panels and the quantification panels clearer. The text changes have been shown in blue in the “Related Manuscript” file of the Article text.

Please also respond to additional points raised by individual reviewers below.Reviewer #1 (Recommendations for the authors):1. One of the most interesting findings in the manuscript is the potential relationship between CAP-G2 C terminal region and CAP-D2. The authors speculate that phosphorylation could regulate this relationship. Could the authors test this in their assay?

We assume that the reviewer meant the relationship between CAP-D3 C-terminal region and CAP-G2 (and NOT “between CAP-G2 C terminal region and CAP-D2”). We also find that the potential, functional relationship between the CAP-D3 C-tail and CAP-G2 is of great interest. Although we provided evidence that at least two sites, T1415 and S1474, are phosphorylated in a mitotic extract (Figure 4—figure supplement 1B), the CAP-D3 C-tail contains 9 additional CDK consensus sites and 33 other serines/threonines. Identification of key phosphorylation sites in the D3 C-tail that might affect its physical and functional interaction with CAP-G2 would require a substantial amount of work and is clearly beyond the scope of the current manuscript. We will address this very important question in future studies.

2. How do the levels of condensin on chromosomes in this assay compare to the levels in vivo? Could the effects observed be due to greatly elevated levels?

We are afraid that we do not fully understand this comment, but interpret it to mean that the reviewer questions the relatively high concentrations of the recombinant complexes added back into the extracts. If so, please see our response to comment #4 in the Essential Revision.

Reviewer #2 (Recommendations for the authors):1) Figure 6 doesn't seem to add much new insight, and it is odd that the authors use *Xenopus* sperm in this figure only and don't explain their motivation for this. Thus, I feel it should either be removed from the paper or be moved to supplementary info.

The reason for the use of *Xenopus* sperm nuclei in this assay has been described in our response to comment #2 in the Essential Revision. The following sentence has been added to the revised manuscript.

Lines 292-294:

“… chromatin remodelers (*Xenopus* sperm nuclei were used as the substrate in this assay because no reproducible protocol for Npm2-mediated swelling of mouse sperm nuclei was available).”

It should also be emphasized that the extract-free assay nicely complements the extract-based add-back assay (also see our response to comment #3 in the Essential Revisions). For instance, the reviewers expressed their concern that residual levels of endogenous subunits remaining in the immunodepleted extracts might incorporate into the recombinant complexes, thereby causing unforeseeable effects in the add-back assay. Npm2-treated chromatin binding assay, which does not use *Xenopus* egg extracts, can recapitulate part of the activities of the recombinant complexes (i.e., chromatin loading and resulting chromatin compaction), thereby helping to rule out the reviewers’ concern. For this reason, we wish to keep Figure 6 as one of the main figures. To further clarify our motivation for the addition of Figure 6 to the current manuscript, we have modified the text in the revised manuscript as follows:

Lines 281-284:

“However, the egg extracts contain a number of activities that modify or potentially supplement the activities of the recombinant condensin II complex. In an attempt to fill

the gap in technical approaches between the extract-based assay and other biochemical/biophysical assays (Kong et al., 2020), we next wished to set up a simple chromatin-binding assay that does not utilize the extracts.”

2) The paper would be significantly enhanced with further mechanistic insights into the repressive function of the D3 tail. This could be achieved through phospho-mimetic mutations, and/or through the addition of excess D3 tail to the D3-dC condition to see if it can be inhibited in trans.

This constructive comment is fully appreciated. We do think that exploring the potential phospho-regulation of condensin II mediated by the D3 C-tail is a very important direction in the future, but such an effort is beyond the scope of the current manuscript. See our response to comment #1 of Reviewer #1.

3) The cartoon in Figure 1D seems more appropriate for the Discussion and Figure 7. To my knowledge, the authors do not provide direct evidence for the loops and axis featured in 1D, which would require Hi-C experiments.

We have removed the cartoon from the revised manuscript.

4) In Figure 1D, for the quantification of the line scans, it would be useful to have chromosomes assembled in WT extracts included so that the reader can be sure of the differences between WT chromosome morphology and "chenille" morphology shown in the representative images are borne out quantitatively.

This constructive comment is fully appreciated. We have replaced the line scans in Figure 1D with a re-quantified version that includes the line scans of the chromosomes assembled in the Δmock ( = WT) extract. The measurements for the new line scans have also been included in Figure 1-source data 1.

5) In the ATP-hydrolysis assays in Figure 3, the authors show that ∆G2(WT) mutants have increased ATP activity compared to holo(WT), both with and without dsDNA as a substrate. Given prior work from the lab demonstrating that condensin I goes through continuous cycles of ATP-hydrolysis-dependent turnover in *Xenopus* egg extracts and that transition state mutations slow this turnover rate, it would be interesting to investigate the kinetics of ∆G2(WT) condensin II turnover. If enhanced ATP-hydrolysis were to be a causal factor in the high chromosomal accumulation of condensin II in ∆G2(WT) mutants (which the authors are careful not to claim outright), then we might expect to see an increased rate of condensin II turnover in ∆G2(WT) cond II compared to holo(WT) cond II as well. Such a result would strengthen the possibility of a mechanistic link between CAP-G2's negative regulation of ATP hydrolysis and its negative regulation of condensin II loading on chromosomes. This could be easily tested with the type of two-color assays this lab has previously performed on condensin I mutants.

We thank the reviewer for the recognition of a technique that was previously reported by our group (Kinoshita et al., 2015). However, we wish to remind the reviewer that the two-color assays require additional construction, expression and purification of the corresponding wild-type and mutant complexes. Although the suggested experiments are beyond the scope of the current study, we expect that they will produce useful information in the future.

6) Also in regard to the ATP-hydrolysis assay in Figure 3, it would be worthwhile to include ∆D3(WT) and ∆H2(WT) mutants for completeness. For example, maybe the enhanced ATP-hydrolysis ability of ∆G(WT) isn't specific to CAP-G function, but is instead a result related to sterics, wherein deletion of any of the condensin non-SMC subunits allows more room for ATP to bind and undergo hydrolysis. Demonstrating that ∆D3(WT) does not have increased ATP-hydrolysis activity compared to holo(WT) would eliminate this possibility and further support the notion of CAP-D3 playing a positive role in the chromosomal accumulation of condensin II and axis formation.

We read the second sentence of this comment as “…maybe the enhanced ATP-hydrolysis ability of ∆G2(W’) isn't specific to CAP-G2 function”. That said, we are afraid the reviewer tries to overinterpret our results. In the current manuscript, we do not claim that the enhanced ATPase rate of DG2(WT) is directly coupled to its enhanced association with chromosomes. We simply say that “CAP-G2 negatively regulates the ATPase activity of condensin II as well as its ability to associate with chromosomes”. In fact, we also show that DG2(SMC-TR) has a negligible level of ATPase activity (Figure 3C; p = < 0.001 to holo[WT]) but displays a higher level of chromosome association than holo(WT) (Figure 3A and B; p = < 0.001). We believe that the suggested experiment is not necessary to further substantiate the major conclusions reported in the current manuscript.

7) The authors write that at 200nM, the structures formed by holo(D3-dC) "were no longer chenille-like," but instead "reminiscent of rod-shaped chromosomes produced by condensin I". It does appear that the chromosome axes are better defined in this condition, as confirmed by the line scans in Figure 4C. However, in order to justify the comparison, the authors should include WT chromosomes both in the representative images in 4B and in the line scans in 4C. After all, the difference in the shape of the normalized DAPI intensity distribution (Figure 4C) is not enormous (and there is no statistical analysis of it), so having the WT distribution for comparison would allow the reader to evaluate how complete the rescue actually is. There should be quantification of the ∆G2(WT) vs ∆G2(D3-dC) DAPI images shown in Figure 4D. Figure 4 Figure Supplement 2C has line scans of the relative DAPI intensity distributions of ∆G2(WT) vs WT and ∆G2(D3dC) compared to WT, but the authors need to include the normalized DAPI intensity distributions of ∆G2(WT) vs ∆G(D3-dC) in Figure 4 since these are the conditions being directly compared in Figure 4D and the authors claim that the morphologies are the same at 200nm (i.e. ∆G2(D3-dC) and ∆G2(WT) are both chenille-like). The CAP-H2 intensities should also be included, meaning Figure 4 Figure Supplement 2 B should be moved to Figure 4 (and the 200nm condition should be quantified).

We wrote that the chromosomes formed by holo(D3-dC) addition at 200 nM were “reminiscent of rod-shaped chromosomes produced by condensin I”. This was based on the observation that the fuzzy chromatin observed in chenille-like chromosomes is more concentrated in the axial center in the holo(D3-dC) chromosomes. We have chosen the modest expression in the current manuscript because we thought that a direct comparison between condensin II holo(D3-dC) and condensin I would require a more detailed understanding of the mechanism of action of holo(D3-dC) and the suppressive function of the D3 C-tail. We fully appreciate that this is a very important issue that needs to be thoroughly addressed in future studies. Such efforts will help to further dissect the similarities and differences between condensin I and condensin II in terms of both their mechanisms of action and regulation.

In our view, the current format of the presentation can be used to illustrate the similarities between ∆G2(WT) and ∆G2(D3-dC) by using holo(WT) as a control. We have made available DAPI and hCAP-H2 measurements of all complexes and conditions in Figure 4, including the hCAP-H2 lines scans for ∆G2(D3-dC) at 200 nM, as source data (Figure 4-source data 1) submitted with the manuscript.

Line 1197-1198:

“Figure 4-source data 1. Microsoft excel of non-normalized data of all conditions corresponding to Figure 4 and Figure 4—figure supplement 2.”

8) In Figure 4 figure supplement 1C, the authors should add ∆G2(D3-dC) mutants to their analysis. ∆G2(D3-dC) mutants are unable to recapitulate the chromosome compaction activity of holo(D3-dC) mutants. It would be natural then, to see if the combination of ∆G2 and D3-dC mutants has any impact on ATP hydrolysis – if combining these mutations lowers the ATP-hydrolysis capacity of the resulting complex, then this would add further insight into why this complex can't match holo(D3-dC) chromosome compaction ability as well as the nature of the "cryptic" function of CAP-G2 exposed by C-terminus deletion.

This comment is related to comment #6 above (see our response to comment #6, too). Although our findings that DG2(WT) and holo(D3-dC) display increased ATPase rates are very illuminating, we consider that a comparison of the ATPase rates between the two mutant complexes would be insufficient to interpret the phenotypic difference observed in the add-back assay. For this reason, we do not think that an additional measurement of the ATPase rate of ∆G2(D3-dC) would give us valuable information. In the future, we wish to address the functional differences among these mutant complexes using different assays including loop extrusion assays.

Reviewer #3 (Recommendations for the authors):1. The quantification of chromosome shape (DNA and condensin distribution across the chromatid) is very useful. However, a direct comparison between conditions incl. wt extracts is difficult as the data is distributed over different graphs. A unifying graph would be helpful.

We assume that the reviewer refers to the multiple graphs presented in Figure 4—figure supplement 2. Due to the large number of different complexes and different concentrations tested, we find it difficult to make a unified graph. We believe that the current compilation of graphs communicates the key conclusions of the manuscript.

2. To facilitate comparison across different model systems, please mention ortholog names of CAP-G and -D subunits in the introduction.

Please note that the names of the budding yeast orthologs (Brn1/CAP-H… Ycg1/CAP-G…) were included in the Discussion of the original manuscript. In the revised manuscript, we have added them in the Introduction, too.

Lines 57-58:

“CAP-H/Brn1… CAP-D2/Ycs4… CAP-G/Ycg1…”

3. Abstract: 'how condensin II might work has remained largely unexplored'. I think this statement is not correct (and not needed) as the basic mechanism is understood, but the regulation is much more unexplored.

The reviewer says that the basic mechanism of condensin II is already understood. We respectfully disagree with this statement. To our knowledge, only two studies have attempted, so far, to test the molecular activities of condensin II using its purified fraction (Houlard et al., 2022; Kong et al., 2020), and our current understanding of the mechanism of condensin II is still far from complete. Moreover, compared to the long history of analyses of condensin I, only a limited set of biochemical/biophysical assays has been applied for the analyses of condensin II. We do agree with the reviewer, however, that the regulation of condensin II is less explored. In the revised manuscript, the abstract text has been modified as follows:

Lines 31-32:

“Although the mechanism of action and regulation of condensin I have been studied extensively, our corresponding knowledge of condensin II remains very limited.”